# Synapse maintenance is impacted by ATAT-2 tubulin acetyltransferase activity and the RPM-1 signaling hub

Melissa A Borgen, Andrew C Giles, Dandan Wang, Brock Grill*

Department of Neuroscience, The Scripps Research Institute, Jupiter, United States

**Abstract** Synapse formation is comprised of target cell recognition, synapse assembly, and synapse maintenance. Maintaining established synaptic connections is essential for generating functional circuitry and synapse instability is a hallmark of neurodegenerative disease. While many molecules impact synapse formation generally, we know little about molecules that affect synapse maintenance in vivo. Using genetics and developmental time course analysis in *C.elegans*, we show that the α-tubulin acetyltransferase ATAT-2 and the signaling hub RPM-1 are required presynaptically to maintain stable synapses. Importantly, the enzymatic acetyltransferase activity of ATAT-2 is required for synapse maintenance. Our analysis revealed that RPM-1 is a hub in a genetic network composed of ATAT-2, PTRN-1 and DLK-1. In this network, ATAT-2 functions independent of the DLK-1 MAPK and likely acts downstream of RPM-1. Thus, our study reveals an important role for tubulin acetyltransferase activity in presynaptic maintenance, which occurs via the RPM-1/ATAT-2 pathway.

DOI: https://doi.org/10.7554/eLife.44040.001

*For correspondence:
bgrill@scripps.edu

Competing interests: The authors declare that no competing interests exist.

## Introduction

Synapse formation is comprised of several steps including target recognition, synapse assembly and synapse maintenance (*Chia et al., 2013*; *Jin and Garner, 2008*). Synapse maintenance, also referred to as synapse stability, is required to complete the synapse formation process, and is also important for maintaining circuitry and allowing plasticity throughout an animal's life (*Lin and Koleske, 2010*). Indeed, increasing evidence indicates synapse instability is a hallmark of many neurodegenerative diseases, including Alzheimer's disease (*Lin and Koleske, 2010*; *Selkoe, 2002*; *Spires-Jones and Hyman, 2014*). Understanding how synapse maintenance influences nervous system development, plasticity and disease will require far greater knowledge of the molecules and signaling networks that regulate this process.

Previous genetic studies have been invaluable for informing our understanding of how synapse maintenance is regulated in vivo. At the fly neuromuscular junction (NMJ), genes encoding regulators of the microtubule cytoskeleton, such as Dynactin and Ankyrin, are crucial for maintaining the presynaptic terminal (*Eaton et al., 2002*; *Pielage et al., 2008*). Spectrin, a scaffold that links cell adhesion with the microtubule cytoskeleton, is also important for NMJ synapse maintenance (*Massaro et al., 2009*; *Pielage et al., 2005*).

In *C. elegans*, mechanosensory neurons that form glutamatergic neuron-neuron synapses (reminiscent of mammalian central synapses) have proven particularly valuable for understanding the molecular and genetic underpinnings of synapse maintenance. For instance, pharmacological and genetic perturbation of microtubules impairs presynaptic bouton maintenance in these cells (*Chen et al., 2014*). Genetic screens using mechanosensory neurons revealed that the microtubule minus-end binding protein PTRN-1/CAMSAP and the actin binding protein ZYX-1 are required for synapse maintenance (*Luo et al., 2014*; *Marcette et al., 2014*; *Richardson et al., 2014*). Thus,

studies from both flies and worms emphasize the power genetic model systems wield in identifying molecules, and potentially unraveling entire signaling networks, that are required for synapse maintenance.

While increasing evidence has linked genetic perturbation of the microtubule cytoskeleton with synapse deterioration, it remains unknown whether mutants that affect post-translational modification of microtubules, such as acetylation, affect synapse maintenance. Two α-tubulin acetyltransferases, MEC-17 and ATAT-2, were identified in *C. elegans*. MEC-17 and ATAT-2 function via enzymatic acetyltransferase activity and non-enzymatic mechanisms to regulate microtubule structure, touch sensation, axon polarity and axon degeneration in mechanosensory neurons (*Akella et al., 2010*; *Neumann and Hilliard, 2014*; *Shida et al., 2010*; *Topalidou et al., 2012*). Despite this prior work, it remains unknown if α-tubulin acetyltransferases affect the synapse, and in particular synapse maintenance, in any system. The importance of addressing this question is highlighted by evidence that altered α-tubulin acetylation is associated with neurodegenerative diseases, such as Alzheimer's and Parkinson's disease (*Godena et al., 2014*; *Govindarajan et al., 2013*; *Hempen and Brion, 1996*; *Pellegrini et al., 2017*).

The Pam/Highwire/RPM-1 (PHR) proteins, including *C. elegans* RPM-1, are enormous signaling hubs that also have ubiquitin ligase activity (*Grill et al., 2016*). PHR proteins are important regulators of neuronal development with conserved roles in synapse formation (*Bloom et al., 2007*; *Schaefer et al., 2000*; *Wan et al., 2000*; *Zhen et al., 2000*), axon guidance (*Bloom et al., 2007*; *Lewcock et al., 2007*; *Park and Rongo, 2018*) and axon termination (*Borgen et al., 2017b*; *Feoktistov and Herman, 2016*; *Schaefer et al., 2000*). PHR proteins regulate synapse formation at NMJs (*Bloom et al., 2007*; *Wan et al., 2000*; *Zhen et al., 2000*) and glutamatergic neuron-neuron synapses formed by *C. elegans* mechanosensory neurons (*Schaefer et al., 2000*). At present, it is unclear whether PHR proteins impact synapse formation by regulating synapse assembly or maintenance. Furthermore, PHR protein signaling can influence microtubules in the context of axon guidance and termination (*Borgen et al., 2017b*; *Hendricks and Jesuthasan, 2009*; *Lewcock et al., 2007*). However, whether there is a functional genetic relationship between PHR proteins and tubulin acetyltransferases and, if so, how this influences synapse formation and maintenance remains unknown.

Here, we use developmental time course analysis, genetics and pharmacology to show that the α-tubulin acetyltransferase ATAT-2 regulates synapse maintenance in mechanosensory neurons, and does so via its enzymatic acetyltransferase activity. Moreover, ATAT-2 functions in a pathway with RPM-1 to regulate presynaptic maintenance in mechanosensory neurons, as well as behavioral habituation to repeated gentle touch. Genetic analysis indicates that RPM-1 is a hub in a network containing ATAT-2, PTRN-1 and DLK-1. Importantly, the RPM-1/ATAT-2 pathway represents a mechanism that functions independent of DLK-1 to regulate synapse maintenance. Overall, our findings not only reveal a novel, functional role for ATAT-2, but place it within an RPM-1 signaling network that is required for synapse maintenance.

## Results

### RPM-1 regulates presynaptic bouton maintenance during development

*C. elegans* has two PLM mechanosensory neurons, each of which has a single primary axon that extends a collateral branch to form chemical synapses (*Figure 1A,B*). Similar to neurons in the mammalian central nervous system, *C. elegans* mechanosensory neurons form glutamatergic, neuron-neuron connections (*Chalfie et al., 1985*; *Lee et al., 1999*). Previous work showed RPM-1, the *C. elegans* PHR protein, is an important regulator of PLM neuron synapse formation (*Grill et al., 2016*; *Schaefer et al., 2000*). In adult animals, the PLM neurons of *rpm-1* mutants lack both the collateral branch and chemical synapses (*Figure 1B*). At present, it is uncertain whether this defect arises from failed synapse assembly or impaired synapse maintenance. To address this, we began by revisiting time-course analysis of PLM presynaptic bouton development in wild-type (wt) animals and *rpm-1* mutants. When L1 larvae hatch, the primary PLM axon has extended, but the synaptic branch is absent (*Figure 1B,C*). By 6 hr post-hatch (PH), wt animals have formed morphological presynaptic boutons (*Figure 1B,C*). In contrast, synaptic bouton development is delayed in *rpm-1* mutants at 6 hours PH. Despite this initial delay, *rpm-1* mutants eventually form synaptic boutons, which are

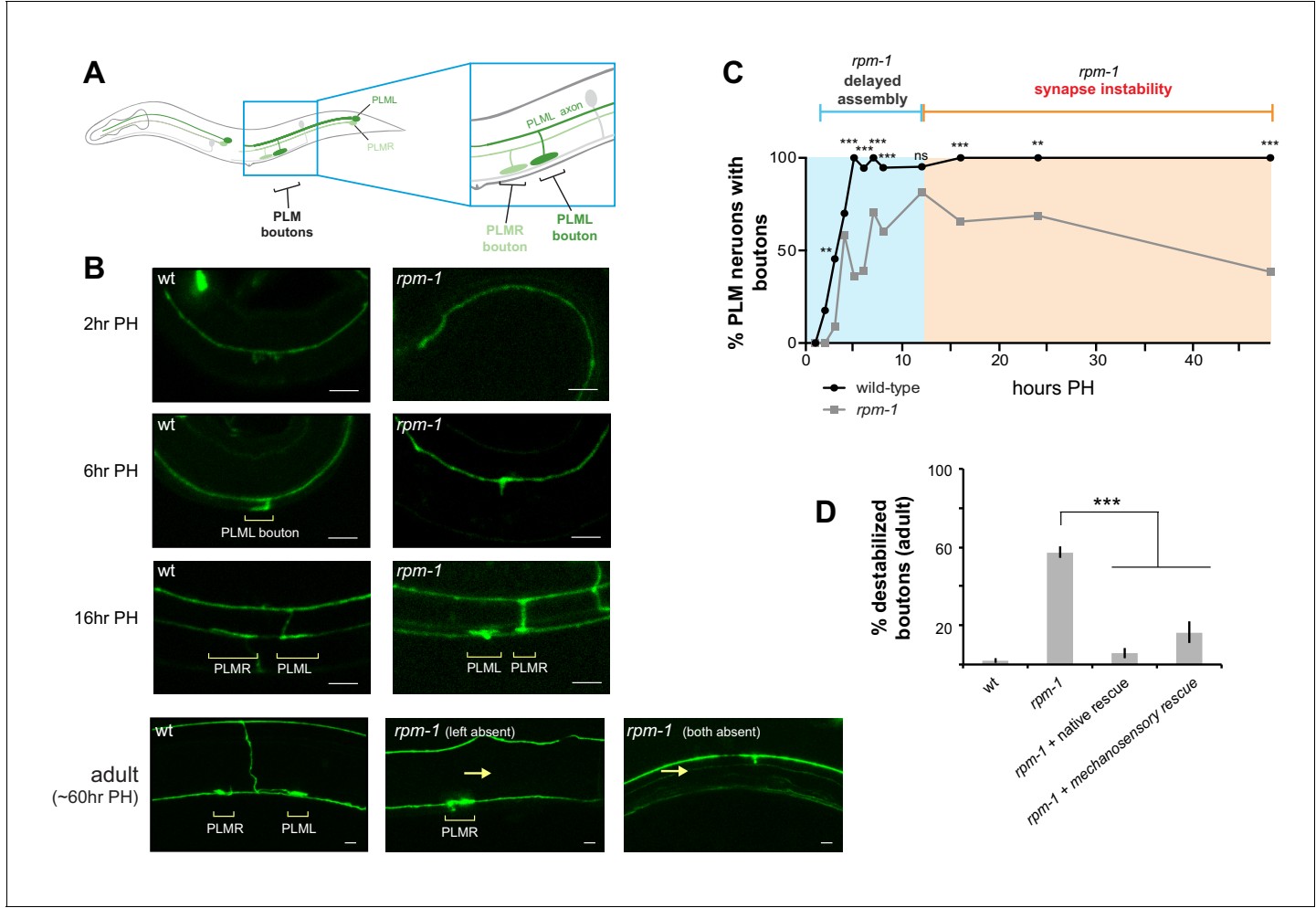

**Figure 1.** RPM-1 functions cell autonomously in mechanosensory neurons to regulate presynaptic bouton maintenance. (a) Schematic highlighting location of collateral synaptic branch and chemical synapses in PLM mechanosensory neurons. (b) Confocal images at different developmental time points showing presynaptic boutons of PLM neurons are delayed in formation and destabilize in *rpm-1* mutants. Note brackets denote PLM presynaptic boutons (PLML and PLMR) and arrows highlight synaptic branch retraction. Note that in confocal images at 16 hr and 60 hr PH, PLML synaptic branch is out of focal plane but bouton is visible. (c) Developmental time course showing synaptic boutons in PLM neurons of *rpm-1* mutants are delayed in formation but reach normal levels by 16 hr PH (blue). Subsequently, *rpm-1* mutant boutons are progressively lost over time (orange). (d) Quantitation showing bouton maintenance defects in adult *rpm-1* mutants, and rescue with transgenic RPM-1 expressed using native *rpm-1* promoter or mechanosensory neuron promoter. Significance tested using Fisher's exact test for c, and Student's *t*-test with Bonferroni correction for d. ***p<0.001, **p<0.01 and ns = not significant (p>0.05).

DOI: https://doi.org/10.7554/eLife.44040.002

readily observable at 12 hr PH (*Figure 1B,C*). Between 12 and 48 hr PH, boutons destabilize and the synaptic branch retracts in *rpm-1* mutants to phenotypic levels characteristic of adult *rpm-1* mutants (*Figure 1B,C*). Transgenic rescue showed expression of RPM-1 using the native promoter or a mechanosensory neuron promoter rescued these defects (*Figure 1D*). These results indicate that RPM-1 primarily regulates presynaptic bouton maintenance by functioning cell autonomously in mechanosensory neurons. Notably, our results differ from a prior study that suggested synaptic branch defects in *rpm-1* mutants arise primarily from impaired initial bouton formation with a relatively small frequency of bouton loss over time (*Schaefer et al., 2000*).

## RPM-1 is localized to presynaptic terminals during development

To assess RPM-1 localization during development, we transgenically expressed RPM-1::GFP in mechanosensory neurons along with tdTOMATO as a cell fill (*Figure 2*). RPM-1 was observed at

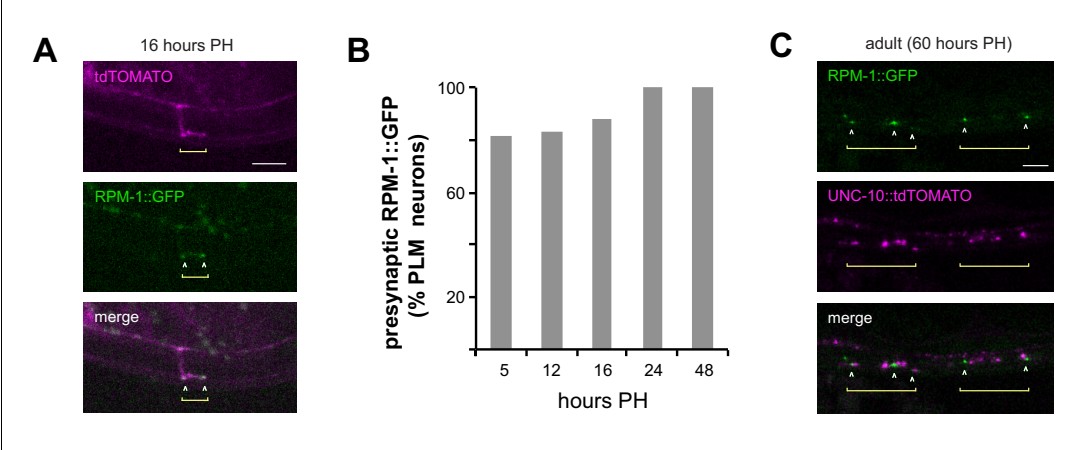

**Figure 2.** RPM-1 localizes to presynaptic terminals of developing and adult mechanosensory neurons. (**a**) Confocal images showing RPM-1::GFP localized at presynaptic boutons of PLM neuron at 16 hr PH. tdTOMATO shows PLM axon and presynaptic terminal morphology. (**b**) Quantitation of RPM-1::GFP presynaptic localization in PLM neurons at different times in development. (**c**) Confocal image showing RPM-1 localized to periactive zones adjacent to active zone marker UNC-10::tdTOMATO in adults.
DOI: https://doi.org/10.7554/eLife.44040.003

presynaptic boutons during synapse development (*Figure 2A*). RPM-1 localization to presynaptic boutons occurred as early as 5 hr PH, when boutons are first consistently present, and was observed through adulthood (*Figure 2B,C*).

To confirm RPM-1 is localized at presynaptic terminals, we used confocal microscopy to evaluate localization of RPM-1 and the active zone component UNC-10/RIM (*Koushika et al., 2001*). In adult animals, RPM-1 localized directly adjacent to UNC-10 at presynaptic terminals (*Figure 2C*). This indicates RPM-1 localizes to the periactive zone of presynaptic terminals in mechanosensory neurons. Our observation is consistent with prior studies that examined RPM-1 localization in motor neurons (*Abrams et al., 2008*).

Collectively, these results support several conclusions. 1) RPM-1 is localized to presynaptic terminals, which is consistent with its role in presynaptic maintenance. 2) Our observation that RPM-1 is present early in the synapse formation process and persists into adulthood suggests signaling that affects presynaptic maintenance could be initiated relatively early in the synapse formation process. 3) Localization of RPM-1 to presynaptic terminals early in development is consistent with delayed presynaptic bouton formation in *rpm-1* mutants.

### *rpm-1* mutants assemble synapses prior to synapse destabilization

Defects in presynaptic bouton morphology and retraction of the synaptic branch suggested *rpm-1* mutants have synapse maintenance defects (*Figure 1*). Failed synapse maintenance could arise because synapses deteriorate or because the synaptic branch successfully extends to postsynaptic target neurons, but fails to properly assemble presynaptic components.

To initially test this, we evaluated a synaptic vesicle marker, RAB-3 (*Luo et al., 2014*; *Nonet et al., 1997*). Using integrated transgenes expressing RAB-3::GFP and RFP cell fill in mechanosensory neurons, we evaluated several developmental time points. In wt animals, synaptic boutons began to form by 3 hr PH, and we observed RAB-3::GFP accumulation at presynaptic terminals even at this early time point in synapse assembly (*Figure 3A*). We also noted that RAB-3 accumulated at the axon point where the synaptic branch initially descends (*Figure 3A*). At 12 and 24 hr PH, RAB-3 was enriched at presynaptic terminals (*Figure 3A*). In *rpm-1* mutants, we observed accumulation of RAB-3 at presynaptic terminals by 12 hours PH (*Figure 3B*). In contrast to wt animals, synapses destabilize in *rpm-1* mutants with synaptic branches retracting by 24 hours PH and RAB-3 present only in the primary axon (*Figure 3B*).

Quantitation across a wider range of developmental time points showed that in wt animals presynaptic bouton frequency is maximal at 12 hr PH, and RAB-3 accumulates in every bouton (*Figure 3C*). RAB-3 accumulation at presynaptic terminals was maintained across all time points

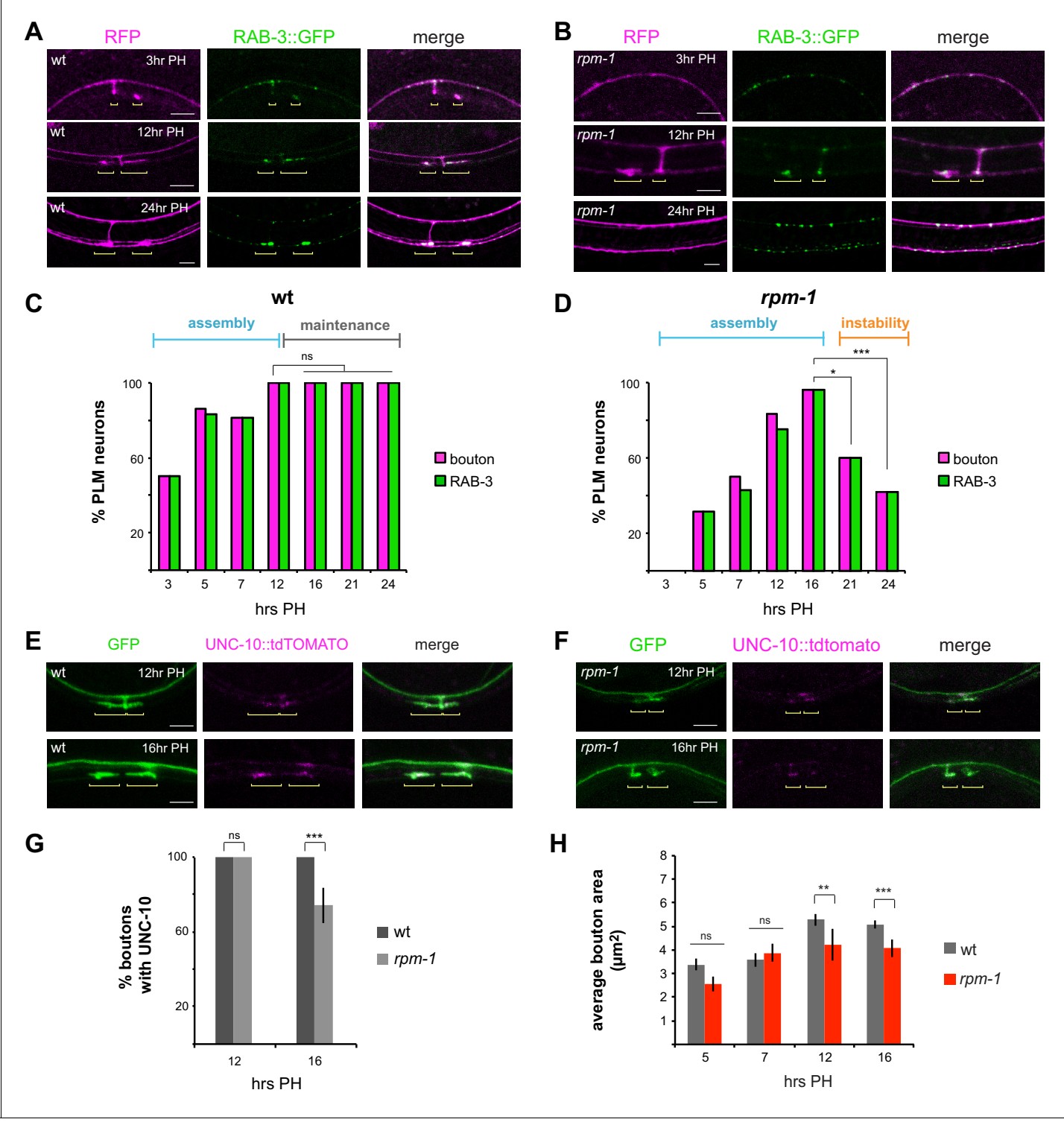

**Figure 3.** RPM-1 regulates synapse maintenance. (a) Confocal images showing synaptic vesicle marker RAB-3 (green) at presynaptic terminals of PLM mechanosensory neurons during development. RFP shows PLM morphology (magenta). Yellow brackets highlight presynaptic terminals of PLML and PLMR neurons. Note one synaptic branch is out of focal plane. (b) *rpm-1* mutants accumulate RAB-3 (green) at presynaptic terminals by 12 hours PH, but presynaptic terminals fail to be maintained leading to synaptic branch retraction by 24 hr PH. (c) Developmental time course of presynaptic boutons and RAB-3::GFP accumulation in wt animals. Full assembly of presynaptic terminals with RAB-3 occurs by 12 hr PH. (d) Developmental time course of presynaptic boutons and RAB-3::GFP in *rpm-1* mutants. Presynaptic assembly with RAB-3 is complete by 16 hours PH, but is not maintained and presynaptic terminals are lost at later time points. (e, f) UNC-10::tdTOMATO marks the active zone and assembles at presynaptic terminals in e) wt and
*Figure 3 continued on next page*

Figure 3 continued

(f) *rpm-1* mutants at critical synapse assembly time points of 12 and 16 hr PH. (g) Quantitation of boutons containing UNC-10. At 12 hr PH, all boutons contain UNC-10 in wt and *rpm-1* mutants. At 16 hr PH, there is a small defect in UNC-10 accumulation at presynaptic terminals of *rpm-1* mutants. (h) Quantitation of bouton area. *rpm-1* boutons are initially the same size as wt boutons (5 and 7 hours PH). *rpm-1* mutants show small decreases in bouton size just prior to synapse loss (12 and 16 hr PH). Significance tested using Fisher's exact test. ***p<0.001, *p<0.05 and ns = not significant.
DOI: https://doi.org/10.7554/eLife.44040.004
The following figure supplement is available for figure 3:

**Figure supplement 1.** SYD-2 active zone marker accumulates in *rpm-1* mutants at 16 hr PH.
DOI: https://doi.org/10.7554/eLife.44040.005

examined (*Figure 3C*). For *rpm-1* mutants, quantitation indicated presynaptic bouton formation is slightly delayed, but all PLM neurons in *rpm-1* mutants have presynaptic boutons containing RAB-3 by 16 hr PH (*Figure 3D*). These results are consistent with *rpm-1* mutants having largely normal, although slightly delayed, synapse assembly. In contrast, presynaptic maintenance is strongly impaired in *rpm-1* mutants with significant, rapid loss of presynaptic boutons and RAB-3 by 21 hr PH, and further reductions by 24 hours PH (*Figure 3D*). Notably, presynaptic terminals that do not destabilize in *rpm-1* mutants retain RAB-3 (*Figure 3D*).

Next, we tested two presynaptic active zone markers, UNC-10/RIM and SYD-2/Liprin. We focused on 12 and 16 hr PH, as RAB-3 analysis demonstrated these are key time points for assessing completion of synapse assembly (*Figure 3A,C*). In wt animals, UNC-10::tdTOMATO labeled presynaptic terminals at both 12 and 16 hr PH (*Figure 3E*). Likewise, presynaptic terminals of *rpm-1* mutants contained UNC-10 at 12 and 16 hr PH (*Figure 3F*). Quantitation indicated that UNC-10 is at all presynaptic terminals in *rpm-1* mutants at 12 hours PH, and the majority of terminals at 16 hours PH (*Figure 3G*). Similar results occurred with mScarlet::SYD-2 (*Figure 3—figure supplement 1*). Interestingly, a small but significant decrease in the number of terminals with UNC-10 occurred in *rpm-1* mutants at 16 hr PH compared to wt animals (*Figure 3G*). While a subtle observation, 16 hr PH is a critical time point just prior to synapse destabilization and branch retraction in *rpm-1* mutants. This observation prompted us to also assess presynaptic bouton size over development in *rpm-1* mutants. Bouton size was normal in *rpm-1* mutants at 5 and 7 hr PH (*Figure 3H*). A small, but significant, decrease in presynaptic bouton size emerged in *rpm-1* mutants at 12 and 16 hr PH (*Figure 3H*).

These results with multiple presynaptic markers support several important points. First, our results indicate that synapses assemble in *rpm-1* mutants, and while delayed, this process is largely normal in *rpm-1* mutants. Second, analysis of presynaptic bouton morphology, RAB-3 and UNC-10 indicate synapses rapidly destabilize in the absence of RPM-1. Finally, we observed subtle changes in bouton size and UNC-10 accumulation just prior to synapse destabilization, which suggests these mild presynaptic changes are likely to signal the onset of failed synapse maintenance. Taken as a whole, these results indicate RPM-1 is an important regulator of synapse maintenance in mechanosensory neurons.

## Loss of RPM-1 in combination with pharmacological manipulation of microtubule stability enhances synapse destabilization

We previously showed that RPM-1 signaling affects microtubule stability during growth cone collapse and axon termination (*Borgen et al., 2017b*). Therefore, we wanted to assess how synapse maintenance defects in *rpm-1* mutants are affected by pharmacological manipulation of microtubule stability. Consistent with prior work (*Chen et al., 2014*; *Richardson et al., 2014*), treating wt animals with colchicine, a microtubule-destabilizing drug, resulted in loss of PLM synapses (*Figure 4A*). Treating *rpm-1* mutants with colchicine significantly enhanced synapse maintenance defects (*Figure 4A*). Conversely, treatment of *rpm-1* mutants with the microtubule-stabilizing drug taxol suppressed synapse maintenance defects (*Figure 4B*). These results are consistent with destabilized synapses in *rpm-1* mutants resulting from less stable microtubules.

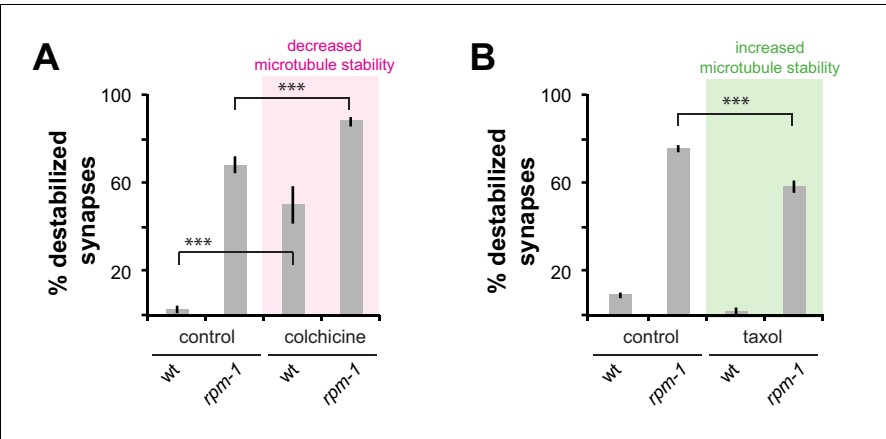

**Figure 4.** Drugs that alter microtubule stability affect synapse maintenance defects in *rpm-1* mutants. (a) Decreasing microtubule stability with colchicine enhances synapse maintenance defects in *rpm-1* mutants. (b) Increasing microtubule stability with taxol suppresses synapse maintenance defects in *rpm-1* mutants. Significance tested using Student's *t*-test with Bonferroni correction. ***p<0.001.
DOI: https://doi.org/10.7554/eLife.44040.006

## ATAT-2 and RPM-1 function in a pathway to regulate synapse maintenance

Next, we wanted to test the genetic relationship between RPM-1 and molecules that affect microtubules. We started by evaluating how null mutants for different microtubule binding proteins and tubulin acetyltransferases affect synapse maintenance. Interestingly, loss of function in *atat-2*, an α-tubulin acetyltransferase, resulted in synapse maintenance defects similar to *rpm-1* mutants in which presynaptic boutons are lost and the synaptic branch is absent (*Figure 5A*). Quantitation indicated theses defects occurred at a moderate but significant frequency in *atat-2* mutants (*Figure 5B*). This observation was confirmed using a second transgenic background (*Figure 5—figure supplement 1*). Destabilized synapses were also observed in mutants for *mec-17*, another α-tubulin acetyltransferase isoform, and the minus-end binding protein *ptrn-1* (*Figure 5B*). However, defects in these mutants occurred at lower frequency than in *atat-2* mutants. Our observation that *ptrn-1* affects PLM synapse maintenance is consistent with a prior study (*Marcette et al., 2014*). Synapse maintenance defects were not observed in *ptl-1/Tau* mutants (*Figure 5B*).

Having tested how different mutants for microtubule binding proteins and tubulin acetyltransferases impact synapse maintenance in PLM mechanosensory neurons, we constructed double mutants with *rpm-1*. Interestingly, *rpm-1; atat-2* double mutants showed a similar frequency of synapse maintenance defects as *rpm-1* single mutants (*Figure 5C*). We validated this result using a second transgenic background (*Figure 5—figure supplement 1*). These results demonstrate that ATAT-2 functions in the same pathway as RPM-1. Similarly, the frequency of synapse maintenance defects was not increased in *rpm-1; ptrn-1* double mutants compared to single mutants, which suggests PTRN-1 and RPM-1 function in the same pathway (*Figure 5C*). We note that this result differs with a prior study that suggested RPM-1 and PTRN-1 function in parallel pathways to regulate synapse formation (*Marcette et al., 2014*). In contrast to outcomes with *atat-2* and *ptrn-1*, the frequency of destabilized synapses was enhanced in *rpm-1; mec-17* and *rpm-1; ptl-1* double mutants (*Figure 5C*). These results indicate that MEC-17 and PTL-1/Tau function in parallel pathways with RPM-1 to regulate synapse maintenance.

Given our observations indicating that RPM-1 functions in the same pathway as both ATAT-2 and PTRN-1, we tested the genetic relationship between *ptrn-1* and *atat-2*. To do so, we evaluated *atat-2; ptrn-1* double mutants. These animals showed strong, significant enhancement of synaptic maintenance defects compared to single mutants (*Figure 5D*). Thus, *ptrn-1* and *atat-2* function in parallel genetic pathways to regulate synapse maintenance.

Previous studies showed one mechanism by which RPM-1 regulates synapse formation is ubiquitination and inhibition of the DLK-1 MAP kinase (*Grill et al., 2007*; *Nakata et al., 2005*; *Yan et al.,*

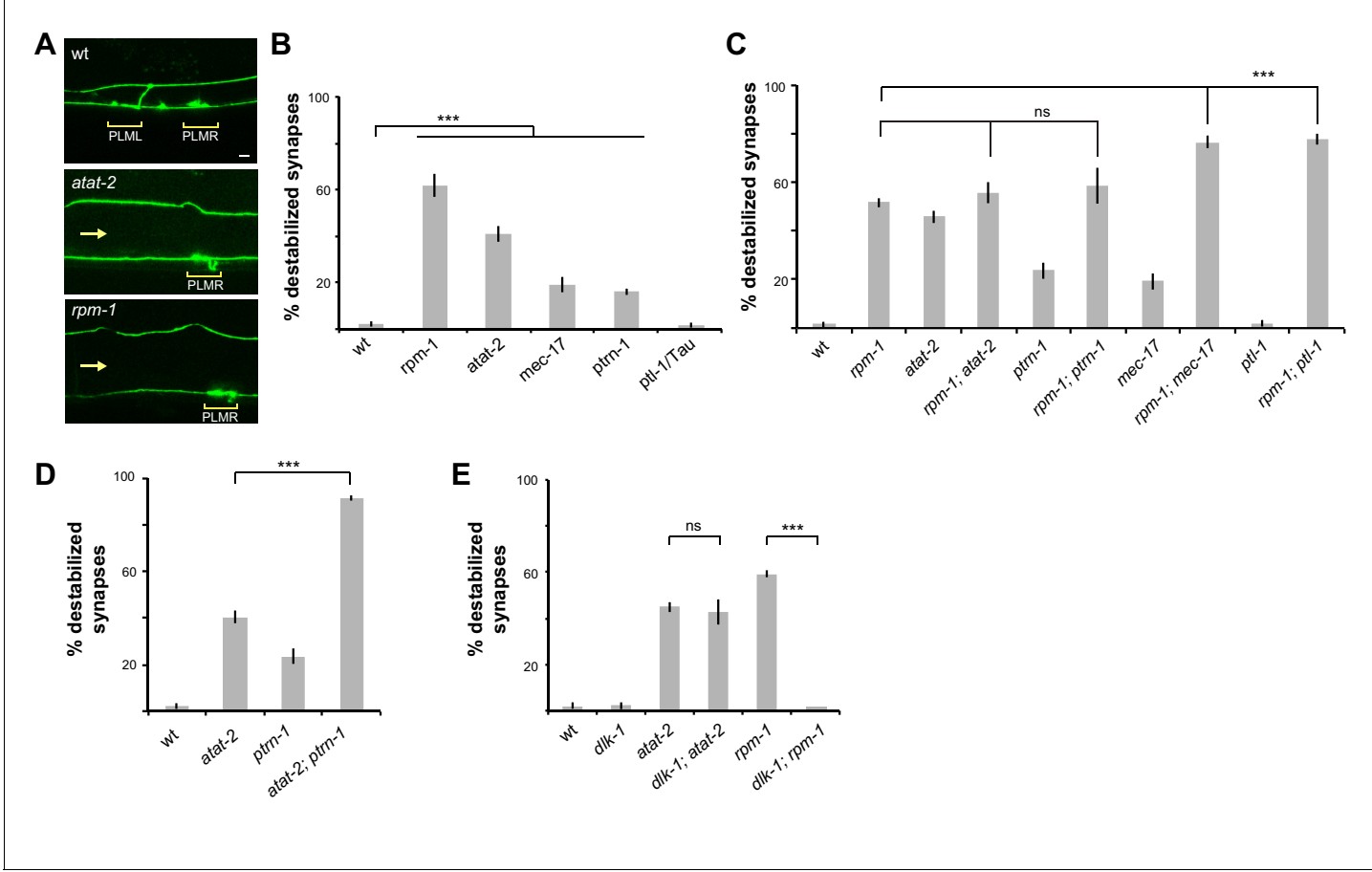

**Figure 5.** Several mutants that affect microtubules interact with *rpm-1* to affect synapse maintenance. (a) Confocal images of presynaptic boutons and synaptic branches in adult PLM neurons. In wt animal, presynaptic boutons from PLML and PLMR are visible. Note one synaptic branch is shown and the other is out of the focal plane. *atat-2* and *rpm-1* mutants lack a synaptic branch and only show PLMR bouton (note loss of PLML, arrow). (b) Quantitation of synapse maintenance defects for indicated genotypes. Note *atat-2* shows higher frequency defects than *ptrn-1* or *mec-17*. (c) Quantitation showing synapse maintenance defects are similar in *rpm-1; atat-2* and *rpm-1; ptrn-1* double mutants compared to *rpm-1* single mutants. In contrast, *rpm-1; mec-17* and *rpm-1; ptl-1* double mutants show enhanced defects compared to *rpm-1* single mutants. (d) Quantitation indicates synapse maintenance defects are enhanced in *atat-2; ptrn-1* double mutants compared to *atat-2* single mutants. (e) Quantitation showing synapse maintenance defects are suppressed in *rpm-1; dlk-1* double mutants, but not *atat-2; dlk-1* double mutants. Significance tested using Student's t-test with Bonferroni correction. \*\*\*p<0.001 and ns = not significant.

DOI: https://doi.org/10.7554/eLife.44040.007

The following figure supplement is available for figure 5:

**Figure supplement 1.** Analysis with a second transgenic background indicates *atat-2* regulates synapse maintenance.
DOI: https://doi.org/10.7554/eLife.44040.008

*2009*). Therefore, we tested the relationship between *dlk-1* and *atat-2*. Consistent with this prior work, synapse maintenance defects were strongly suppressed in *dlk-1; rpm-1* double mutants compared to *rpm-1* single mutants (*Figure 5E*). In contrast, we did not observe suppression of synapse maintenance defects in *dlk-1; atat-2* double mutants compared to *atat-2* single mutants (*Figure 5E*).

To our knowledge, these results show for the first time that the tubulin acetyltransferases ATAT-2 and MEC-17 are required for presynaptic bouton maintenance, with ATAT-2 playing a particularly prominent role. Furthermore, our results demonstrate that ATAT-2 and RPM-1 function in a novel pathway to regulate synapse maintenance, while ATAT-2 functions in parallel to PTRN-1 and independently of DLK-1. The simplest model that explains our findings is that ATAT-2 and DLK-1 are part of a signaling network that is differentially regulated downstream of RPM-1. Because *dlk-1* suppresses *rpm-1* but not *atat-2*, it is particularly likely that ATAT-2 functions downstream of RPM-1. If

ATAT-2 functioned upstream of RPM-1, one would expect suppression of both *rpm-1* and *atat-2* by *dlk-1*, which did not occur.

## ATAT-2 acetyltransferase activity is required for synapse maintenance

Genetic interactions with *rpm-1*, and the loss of presynaptic boutons and synaptic branches in PLM neurons of *atat-2* mutants prompted us to further evaluate if synapse maintenance was impaired in these animals. Indeed, we observed that presynaptic terminals form and accumulate UNC-10/RIM and RAB-3 in *atat-2* mutants at 16 hr PH (*Figure 6—figure supplement 1A,C*). Quantitation indicated that a small defect in UNC-10/RIM accumulation occurred at 16 hours PH in *atat-2* mutants (*Figure 6—figure supplement 1B*). Thus, similar to *rpm-1* mutants, presynaptic assembly is largely normal in *atat-2* mutants with subtle defects in accumulation of UNC-10 occurring at the critical 16-hr PH developmental time point. These results indicate ATAT-2 is affecting synapse maintenance, which is consistent with ATAT-2 functioning in the same pathway as RPM-1.

To further validate synapse maintenance defects in *atat-2* mutants, we performed transgenic rescue experiments. Synapse destabilization in *atat-2* mutants was rescued by expression of ATAT-2 using either the native *atat-2* promoter or a mechanosensory neuron promoter (*Figure 6A,B*). These results demonstrate that ATAT-2 functions cell autonomously in mechanosensory neurons to regulate presynaptic maintenance.

Importantly, rescue experiments allowed us to test whether the enzymatic acetyltransferase activity of ATAT-2, or non-enzymatic mechanisms affect synapse maintenance. To do so, we performed rescue with ATAT-2 (*G125W, G127W*) which lacks acetyltransferase activity (*Topalidou et al., 2012*). Unlike wt ATAT-2, acetyltransferase dead ATAT-2 failed to rescue synapse maintenance defects

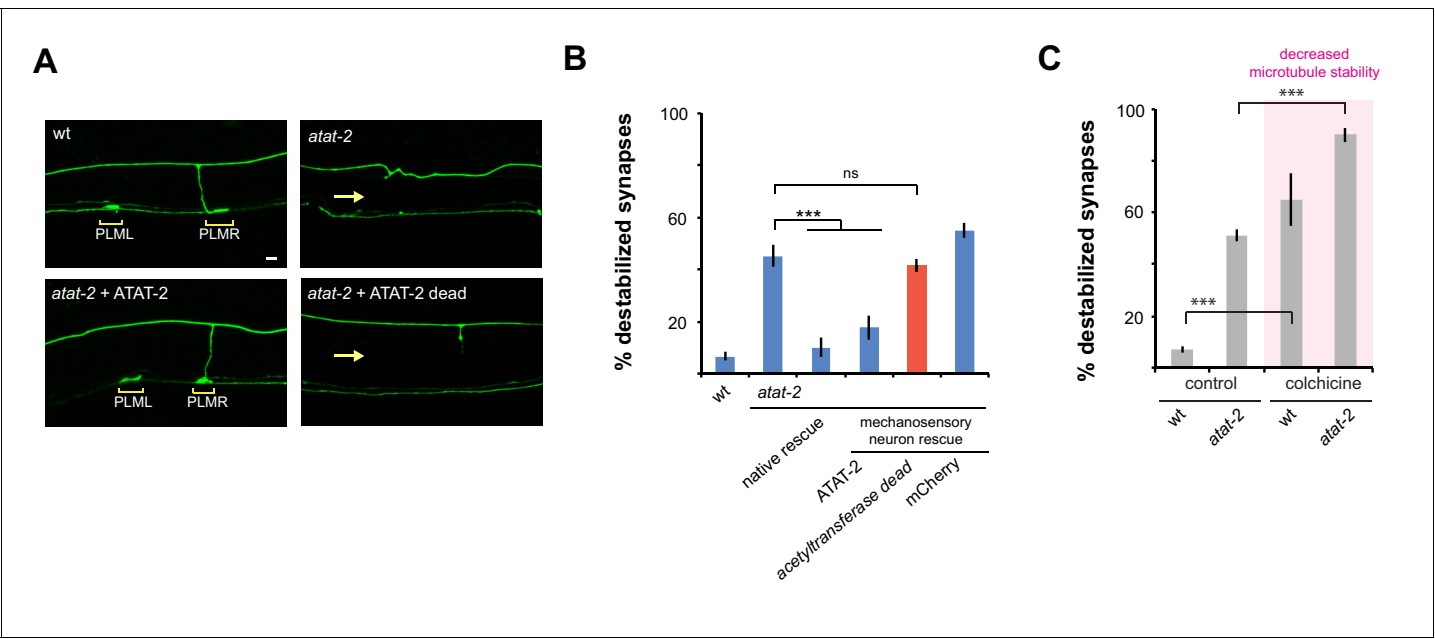

**Figure 6.** ATAT-2 tubulin acetyltransferase activity functions in mechanosensory neurons to regulate presynaptic maintenance. (a) Confocal images of presynaptic boutons and synaptic branches in adult PLM neurons. In the wt animal, presynaptic boutons are shown for both PLML and PLMR neurons (brackets). Note one synaptic branch is shown and other is out of focal plane. *atat-2* mutant lacks presynaptic boutons from both PLML and PLMR, and synaptic branch absent (arrow). Expression of ATAT-2 in mechanosensory neurons rescues defects. ATAT-2 lacking acetyltransferase activity (ATAT-2 dead) fails to rescue (b) Quantitation of synapse maintenance defects for indicated genotypes. Defects in *atat-2* mutants are rescued by using native *atat-2* or mechanosensory neuron promoters to transgenically express ATAT-2. No significant rescue occurs with ATAT-2 lacking acetyltransferase activity. (c) Microtubule destabilizing drug colchicine enhances synapse maintenance defects in *atat-2* mutants. Significance tested using Student's *t*-test with Bonferroni correction. ***p<0.001 and ns = not significant.

DOI: https://doi.org/10.7554/eLife.44040.009

The following figure supplement is available for figure 6:

**Figure supplement 1.** UNC-10/RIM and RAB-3 accumulate at presynaptic terminals of *atat-2* mutants early in development.

DOI: https://doi.org/10.7554/eLife.44040.010

(*Figure 6A,B*). This indicates that ATAT-2 functions via its acetyltransferase activity to regulate synapse maintenance. This finding is consistent with the nature of the *atat-2* allele we used, *ok2415*, which contains a large deletion in the acetyltransferase domain (*Shida et al., 2010*).

To provide further evidence that ATAT-2 influences microtubule stability to affect synapse maintenance, we evaluated how the microtubule destabilizing drug, colchicine, affects defects in *atat-2* mutants. Consistent with ATAT-2 function increasing microtubule stability, synapse maintenance defects caused by *atat-2* (lf) were enhanced by colchicine (*Figure 6C*).

Taken together, these results demonstrate that ATAT-2 α-tubulin acetyltransferase activity regulates synapse maintenance by functioning in mechanosensory neurons. Importantly, this is the first evidence in any system that tubulin acetyltransferase activity is required to maintain synapse stability.

## RPM-1 and ATAT-2 function in a pathway to regulate short-term learning

Our results showed that ATAT-2 and RPM-1 function in a linear pathway to regulate maintenance of chemical synapses in PLM mechanosensory neurons. To test the genetic relationship between *rpm-1* and *atat-2* in a behavioral context, we evaluated habituation to repeated tap stimuli, a form of gentle touch, that is sensed by the mechanosensory neurons.

*C. elegans* respond to tapping the plate they are grown on by reversing their direction of movement. Initial tap sensation is thought to be primarily mediated by electrical gap junction synapses along the primary axon (*Figure 7A*) (*Chalfie et al., 1985*; *Wicks and Rankin, 1995*). Repeated tap stimulus leads to habituation, a simple form of short-term learning in which responses progressively decrease. Tap habituation is influenced by the glutamatergic chemical synapses of mechanosensory neurons (*Crawley et al., 2017*; *Giles et al., 2015*; *Rankin and Wicks, 2000*). RPM-1 functions in mechanosensory neurons to regulate habituation, and habituation defects in *rpm-1* mutants likely result, at least in part, from defects in chemical synapse formation (*Crawley et al., 2017*; *Giles et al., 2015*). Because electrical synapse formation is not impaired in *rpm-1* mutants, they respond normally to initial tap (*Borgen et al., 2017b*; *Giles et al., 2015*; *Meng et al., 2016*).

Consistent with prior studies, wt adult animals habituated to repeated tap with decreased responses over time, while habituation was strongly impaired in *rpm-1* mutants (*Figure 7B*).

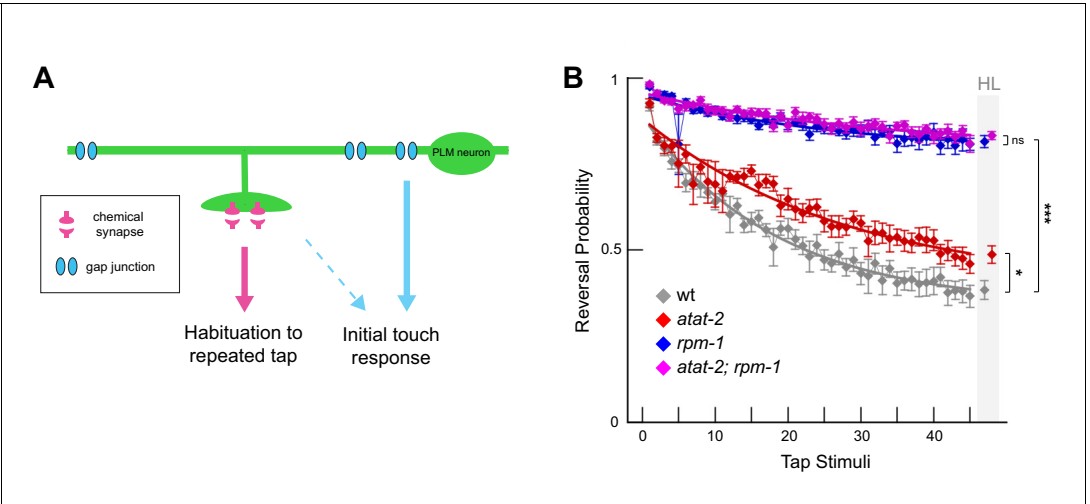

**Figure 7.** Habituation to repeated mechanical stimulation is affected by ATAT-2 and RPM-1. (a) Chemical and electrical synapses in PLM mechanosensory neurons primarily affect habituation to repeated tap stimuli and initial tap sensation, respectively (adapted from *Crawley et al., 2017*). (b) Multi-worm tracker was used to quantitate tap habituation for indicated adult genotypes. Habituation is defective in *atat-2* mutants (red) and *rpm-1* mutants (blue) compared to wt animals. *rpm-1; atat-2* double mutants (magenta) are not significantly different than *rpm-1* single mutants indicating *atat-2* and *rpm-1* function in a linear pathway to regulate habituation. Habituation level (HL) is shaded in grey. Significance assessed by Student's *t*-test with Bonferroni correction. *p<0.05, ***p<0.001 and ns = not significant.
DOI: https://doi.org/10.7554/eLife.44040.011

Consistent with defects in PLM synapse maintenance, habituation was also defective in *atat-2* animals, although defects were less severe than those observed in *rpm-1* animals (*Figure 7B*). *rpm-1; atat-2* double mutants phenocopied *rpm-1* single mutants (*Figure 7B*). These results with whole animal behavior provide further support that RPM-1 and ATAT-2 function in the same pathway.

Prior studies (*Crawley et al., 2017*; *Giles et al., 2015*; *Rankin and Wicks, 2000*) and our results here are all consistent with impaired chemical synapses in mechanosensory neurons affecting habituation to repeated tap, a simple form of short-term learning. However, it is notable that the frequency of synapse maintenance defects in *rpm-1* mutants is only slightly stronger than *atat-2* mutants (*Figure 5B*), while habituation defects are much stronger in *rpm-1* mutants (*Figure 7B*). This might occur because RPM-1 is a signaling hub that regulates several downstream pathways (*Grill et al., 2016*). In contrast, *atat-2* is only known to affect microtubules. Because synapses in *rpm-1* mutants face several insults to signaling compared to *atat-2* mutants, it is possible remaining synapses that do not destabilize could be functionally weaker in *rpm-1* mutants than *atat-2* mutants. This idea is supported by previous studies in Drosophila which have shown that loss of function in the RPM-1 ortholog Highwire results in synapse formation defects and synaptic transmission defects that are mediated by distinct molecular mechanisms (*Borgen et al., 2017a*; *Collins et al., 2006*).

## Discussion

This study breaks new ground on several fronts regarding the molecular and genetic mechanisms that regulate synapse maintenance (*Figure 8*). 1) We provide new evidence that PHR proteins, such as RPM-1, impact synapse formation during development primarily via effects on synapse maintenance. 2) We show for the first time that ATAT-2 regulates presynaptic maintenance, and does so via its α-tubulin acetyltransferase activity. 3) ATAT-2 acts in a novel pathway with RPM-1 that functions cell autonomously to regulate presynaptic maintenance. 4) Extensive genetic analysis revealed RPM-1 is a hub in a signaling network consisting of ATAT-2, PTRN-1 and DLK-1. 5) Finally, our results indicate that ATAT-2 functions independently of DLK-1, and is therefore likely to function downstream of RPM-1 to affect synapse maintenance.

### RPM-1 regulates synapse maintenance during development

Experiments in worms, flies and mice showed that PHR proteins, such as RPM-1, regulate synapse formation (*Bloom et al., 2007*; *Grill et al., 2016*; *Schaefer et al., 2000*; *Wan et al., 2000*; *Zhen et al., 2000*). Important progress has been made in understanding signaling networks regulated by RPM-1 and PHR proteins, which act as both signaling hubs and ubiquitin ligases (*Grill et al., 2016*). Nonetheless, we still lack a clear cellular explanation for why synapse formation is abnormal

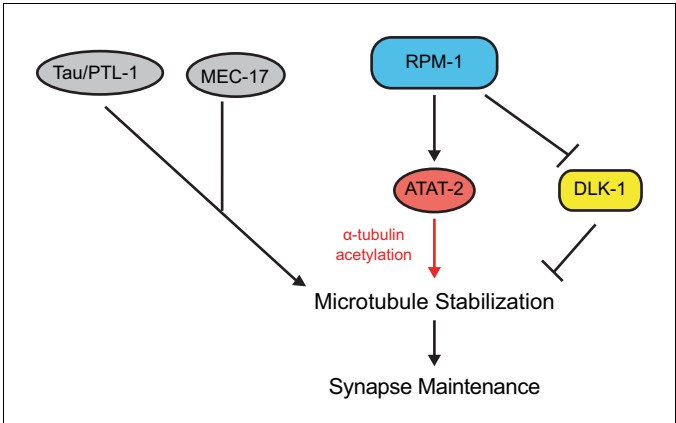

**Figure 8.** ATAT-2 tubulin acetyltransferase activity functions in a pathway with RPM-1 to regulate synapse maintenance. Model summarizing genetic and pharmacological results suggesting RPM-1 functions upstream of ATAT-2 acetyltransferase activity to regulate microtubule stability and synapse maintenance. Outcomes indicate the RPM-1/ATAT-2 pathway functions independently of DLK-1 to regulate synapse maintenance.
DOI: https://doi.org/10.7554/eLife.44040.012

in the absence of PHR proteins. One prior study examined the presence of presynaptic boutons, and hinted that synapse assembly might be the principal defect in *rpm-1* mutants with possible minor defects in synapse maintenance (*Schaefer et al., 2000*). We now expand significantly on this prior work with more extensive developmental time course analysis of bouton morphology, and several presynaptic markers. Our results indicate that synapse formation defects in *rpm-1* mutants result primarily from a failure to maintain synapses (*Figures 1*, *3* and *Figures 3—figure supplement 1*). Consistent with RPM-1 regulating synapse maintenance during development, we observed RPM-1 accumulation at presynaptic terminals during synapse formation through adulthood (*Figure 2*).

In mice, the RPM-1 ortholog Phr1 regulates synapse formation at the NMJ. While it remains unknown if this is due to defects in synapse assembly or maintenance, the Diantonio group observed orphan presynaptic terminals in *Phr1* knockout mice (*Bloom et al., 2007*). Our results suggest these orphan presynaptic terminals could reflect failed synapse maintenance. Consistent with this, studies in flies and mice indicate orphan terminals are a hallmark of destabilizing synapses (*Bhattacharya et al., 2016*; *Eaton et al., 2002*; *Graf et al., 2011*; *Pielage et al., 2008*).

Our findings demonstrate that RPM-1 regulates developmental synapse maintenance, which we consider the final step in synapse formation. This differs from long-term synapse maintenance, which facilitates synapse integrity for months and is regulated by molecules like αLaminin and p190Rho at central synapses, and LRP4 at NMJs (*Barik et al., 2014*; *Kerrisk et al., 2013*; *Lin and Koleske, 2010*; *Omar et al., 2017*). The relationship between regulators of developmental synapse maintenance and long-term synapse maintenance remains unclear and awaits future studies.

## ATAT-2 functions via acetyltransferase activity to regulate synapse maintenance

In *C. elegans,* there are two α-tubulin acetyltransferases, ATAT-2 and MEC-17 (*Akella et al., 2010*; *Shida et al., 2010*). We show here that both molecules affect synapse formation in mechanosensory neurons (*Figure 5*). ATAT-2 is a more prominent player and functions via enzymatic acetyltransferase activity to regulate synapse maintenance (*Figures 5* and *6*). The implications of these findings potentially extend beyond *C. elegans*, as the mammalian acetyltransferase ortholog called Atat1 affects hippocampal development and touch sensation (*Kalebic et al., 2013*; *Kim et al., 2013*; *Morley et al., 2016*). Whether these phenotypes arise from defects in synapse maintenance now becomes an intriguing question.

Our genetic analysis also revealed that ATAT-2 functions in a linear pathway with RPM-1 (*Figure 8*). This was observed in two functional contexts: synapse maintenance in mechanosensory neurons (*Figure 5*) and behavioral habituation mediated by these neurons (*Figure 7*). Consistent with RPM-1 and ATAT-2 acting in a linear pathway, both RPM-1 and ATAT-2 function cell autonomously in mechanosensory neurons to regulate presynaptic maintenance (*Figures 1D* and *6A,B*).

Several observations indicate the RPM-1/ATAT-2 pathway influences microtubule stability to impact synapse maintenance. Treating either *rpm-1* or *atat-2* mutants with colchicine enhanced synapse destabilization defects (*Figures 4A* and *6C*). RPM-1 functions in parallel to other molecules that can stabilize microtubules, such as PLT-1/Tau and MEC-17 (*Figure 5*). Finally, the enzymatic α-tubulin acetyltransferase activity of ATAT-2 was necessary for synapse maintenance (*Figure 6B*).

Our results do not provide definitive evidence for the order of ATAT-2 and RPM-1 within this novel pathway. We attempted to address this with transgenic bypass experiments, but results were inconclusive. Nonetheless, there are several reasons why RPM-1 most likely functions upstream of ATAT-2 (*Figure 8*). First, RPM-1 is a signaling hub and ubiquitin ligase that positively and negatively regulates at least six different downstream signaling pathways (*Grill et al., 2016*). In contrast, ATAT-2 regulates microtubules directly and is not known to regulate signaling events (*Akella et al., 2010*; *Shida et al., 2010*; *Topalidou et al., 2012*). Second, RPM-1 functions in parallel to other molecules that can stabilize microtubules, including MEC-17 and PTL-1/Tau. If RPM-1 were to function downstream of microtubule stability in general, we might expect the same genetic relationship between RPM-1 and all mutants that affect microtubule stability, which did not occur. The final argument is perhaps the most convincing. Our results indicate that ATAT-2, PTRN-1 and DLK-1 function within the RPM-1 signaling network (*Figure 5*). While *dlk-1* can suppress *rpm-1*, it failed to suppress *atat-2*. The simplest explanation for this is a model in which ATAT-2 and DLK-1 have opposing functions with both molecules acting downstream of RPM-1 (*Figure 8*). If ATAT-2 were to function upstream of RPM-1, we would expect *dlk-1* to suppress both *atat-2* and *rpm-1*, which did not occur. Despite

cumulative reasons for favoring the model that RPM-1 functions upstream of ATAT-2, we cannot entirely rule out the alternative possibility.

It is intriguing that the functional genetic relationship between RPM-1 and molecules that affect microtubule stability differs between synapse maintenance in mechanosensory neurons described here (*Figure 8*), and axon termination of the same neurons that occurs in a different anatomical location (*Borgen et al., 2017b*). For example, RPM-1 functions in the same pathway as both ATAT-2 and PTRN-1 to regulate synapse maintenance (*Figure 5C*), but acts in parallel opposing pathways to these molecules to regulate axon termination (*Borgen et al., 2017b*). Further, RPM-1 functions in parallel to PTL-1/Tau to regulate synapse maintenance (*Figure 5C*), but Tau likely inhibits RPM-1 during axon termination (*Borgen et al., 2017b*). Thus, the functional genetic relationship between RPM-1 and regulators of microtubule stability varies with subcellular location and the developmental process in question. These findings in *C. elegans* might explain why studies in fish and mice that analyzed different types of neurons arrived at opposing conclusions about how PHR protein signaling influences microtubule stability (*Hendricks and Jesuthasan, 2009*; *Lewcock et al., 2007*).

Another worthwhile consideration emerges from our findings here, and prior observations about axon termination (*Borgen et al., 2017b*). We now uncover a second example of a functional genetic relationship between RPM-1 and PTL-1/Tau, which has interesting implications given the prominence of Tau in neurodegenerative disease (*Brunden et al., 2009*; *Wang and Mandelkow, 2016*). Likewise, our discovery that RPM-1 functions in a pathway with ATAT-2 acetyltransferase activity to regulate synapse maintenance could have important disease implications, as alterations in α-tubulin acetylation and synapse instability are associated with neurodegenerative diseases (*Godena et al., 2014*; *Govindarajan et al., 2013*; *Hempen and Brion, 1996*; *Lin and Koleske, 2010*; *Pellegrini et al., 2017*). Thus, converging themes from several studies, including this one, suggest it could be informative to test whether the RPM-1/ATAT-2 pathway impacts neurodegenerative disease models.

## Materials and methods

### Genetics and transgenics

*C. elegans* strains were maintained using standard procedures. Alleles used included: *rpm-1 (ju44)*, *mec-17 (ok2109)*, *ptrn-1 (tm5597)*, *ptl-1 (ok621)*, *dlk-1 (ju476)*, and *atat-2 (ok2415)*. Integrated transgenes and extrachromosomal arrays used in this study are as follows: *muIs32* ($P_{mec-7}$GFP), *zdIs5* ($P_{mec-4}$GFP), *bggEx8* ($P_{rpm-1}$RPM-1), *bggEx127* ($P_{mec-3}$RPM-1::GFP), *bggEx141* ($P_{mec-7}$SYD-2::mScarlet), *jsIs973* ($P_{mec-7}$mRFP), *jsIs821* ($P_{mec-7}$RAB-3::GFP), *bggIs28* ($P_{mec-7}$UNC-10::tdTOMATO), and *bggIs34* ($P_{mec-3}$RPM-1::GFP). *jsIs973* and *jsIs821* were kind gifts from Dr. Michael Nonet (Washington University). All alleles were outcrossed a minimum of four times. All double mutants were constructed following standard mating procedures. Genotypes were confirmed by PCR, or sequencing as needed. Primers and PCR conditions are available upon request.

For rescue experiments, transgenic extrachromosomal arrays were constructed by injecting DNA of interest with a coinjection marker, $P_{myo-2}$RFP (2 ng/µl) or $P_{ttx-3}$RFP (50 ng/µL) and pBluescript to reach a total DNA concentration of 100 ng/µL. For all rescue experiments, two or more independently derived transgenic lines were analyzed for a given genotype. *Supplementary file 1* details transgenic extrachromosomal arrays and injection conditions.

### Cloning

For *atat-2* rescues with the native promoter, the *atat-2* locus (including promoter, open reading frame, and 3' UTR) was PCR amplified from N2 genomic DNA. Primer sequences used for cloning are available upon request. For *atat-2* rescue with mechanosensory neuron promoters, *atat-2* cDNA was amplified from *C. elegans* RNA and TOPO cloned into pCR8 Gateway entry vector (Invitrogen) to generate pBG-GY896. pBG-GY896 was recombined into a destination vector containing the *mec-7* promoter, pBG-GY119, to generate pBG-GY897 ($P_{mec-7}$ATAT-2). Site-directed mutagenesis was performed on pBG-GY896 to change two glycine residues into tryptophan (G125W and G127W) resulting in pBG-GY898. Mutation of these conserved glycine residues in the ATAT-2 paralog MEC-17 (G121W and G123W) was previously shown to render MEC-17 catalytically inactive (*Topalidou et al., 2012*). After mutagenesis, pBG-GY898 was recombined with the P*mec-7*

destination vector to yield pBG-GY899 (P$_{mec-7}$ATAT-2 dead). For expression of SYD-2::mScarlet, *syd-2* genomic DNA was cloned from *C. elegans* and TOPO cloned into pCR8 Gateway entry vector to generate pBG-GY699. pBG-GY699 was recombined into a destination vector containing the *mec-7* promoter and a C-terminal mScarlet tag, pBG-GY880, to generate pBG-GY936 (P$_{mec-7}$SYD-2:: mScarlet).

## Developmental analysis of synapses

The transgenic strain *muIs32* (P$_{mec-7}$GFP) was used to label PLM neurons for analysis of synaptic branch and boutons during development. Where indicated, *bggIs28* was used to coexpress the active zone marker UNC-10::tdTOMATO. Transgenic arrays were used to express SYD-2::mScarlet in *muIs32*. For RAB-3::GFP analysis (*jsIs821*), *jsIs973* (P$_{mec-7}$RFP) was used as a cell fill to visualize PLM axon and presynaptic bouton morphology. Synchronized developmental time course analysis was done by collecting freshly hatched L1 larvae and aging animals in hour-long intervals at room temperature (22°C). At indicated time points between 1 and 48 hr post-hatch (PH), animals were mounted in 5 mM levamisole (M9 buffer) on agar pads to score phenotypes. A minimum of 18 PLM neurons were scored at each time point for all markers with the exception of SYD-2::mScarlet which was scored in at least 14 PLM neurons. Phenotypes were scored using epifluorescent microscopy, and all images were acquired using confocal microscopy. Epifluorescent microscopy was done using 100x magnification on a Leica CTR6500 microscope with Leica Application Suite software. Confocal microscopy was done under 63x magnification and 2x zoom factor on a Leica SP8 confocal microscope. Z-stacks were collected (0.5 μm slices for larvae and 1 μm for adults) and maximum intensity projections are shown for each genotype. Image analysis was done using ImageJ software from NIH image (http://rsb.info.nih.gov/ij/).

Synaptic bouton defects were scored by tracking the PLM synaptic branch to the ventral nerve cord. Branches with presynaptic boutons/varicosities were scored as synapses. If no bouton was discernable, even if the branch was present, it was scored as a destabilized synapse. Similar logic was used to evaluate RAB-3::GFP, UNC-10::tdTOMATO and SYD-2::mScarlet at presynaptic terminals.

## RPM-1 localization in PLM neurons

Transgenic *ju44* mutants expressing RPM-1::GFP using the *mec-3* promoter (P$_{mec-3}$RPM-1::GFP) and tdTOMATO using the *mec-7* promoter (P$_{mec-7}$tdTOMATO) were anesthetized with levamisole. Freshly hatched L1 larvae were allowed to age for 5–6 hr at 22°C before imaging. RPM-1::GFP and tdTOMATO were assessed using confocal microscopy and 63x magnification with 2x zoom factor. Adult animals were imaged using 40x magnification and 2x zoom. Increased fluorescence of tdTO-MATO was essential for visualizing PLM morphology in L1 larvae because of their small size, and relatively weak expression of the *mec-7* promoter at this age (data not shown). Periactive localization of RPM-1 at presynaptic terminals was assessed using both RPM-1::GFP and UNC-10::tdTOMATO as an active zone marker. Because UNC-10::tdTOMATO is difficult to observe in early L1 animals due to low expression and small boutons, we performed this analysis using young adult PLM synapses.

## Microtubule pharmacology

For pharmacological manipulation of microtubule stability, taxol (2 μm), colchicine (0.25 mM) or the vehicle DMSO were spread on NGM plates and left overnight. Plates were seeded with OP50 *E. coli*, and 16 hr later 3–5 P0 adults were placed on plates with drugs. F1 progeny developed normally on colchicine or taxol, and synapse maintenance was scored in young adults. Opposing effects of taxol and colchicine on *rpm-1* mutants indicates these drugs are at appropriate concentrations for use in *C. elegans*, which was also shown previously (*Borgen et al., 2017b*).

## Habituation

Tap habituation experiments were performed as described previously with minor modifications (*Giles et al., 2015*). Briefly, age-synchronized animals (~50–100) were cultivated from egg until gravid adult (72–75 hours PH) at 23°C, and assayed on 5 cm NGM plates with 50 μL of *E. coli*. (OP50). Using Multi-Worm Tracker (*Swierczek et al., 2011*), animal behavior was recorded for 550 s.

After the first 100 s, 45 tap stimuli were given with a 10 s inter-stimulus interval. Response to tap was measured by reversal probability (the fraction of animals that reversed their locomotion within 2 s of the tap). For each plate, exponential curves were fit to responses across stimuli, and habituation level (HL) was measured as the value of the fit at the final stimulus. All strains analyzed contained the *muIs32* transgene.

## Statistics

For developmental time course analysis, we used the Fisher's exact test to compare the percentage of PLM neurons lacking synapses in single populations of *rpm-1* and wt at each time point. Data points presented represent the mean of the population. A minimum of 18 PLM neurons were scored at each time point for all markers, except SYD-2::mScarlet which was scored in at least 14 PLM neurons.

For analysis of synapse maintenance defects (% destabilized synapses), we scored a minimum of five independent sets of 20–50 PLM neurons from adult animals for each genotype. For rescue experiments, data was acquired from a minimum of two independent transgenic lines for each genotype. Data shown represents the mean and error bars represent the SEM. Comparisons between genotypes were done using the Student's *t*-test, with Bonferroni correction for the number of comparisons in each experiment.

For habituation, data presented represents the mean and error bars represent the SEM across 12 plates (50–100 animals per plate) tested on three independent days per genotype. Differences were assessed by comparing habituation level using Student's *t*-tests with Bonferroni correction for multiple comparisons.

## Acknowledgements

We thank the *C elegans* knockout consortium for several alleles, and the *C elegans* Genetics Center (NIH Office of Research Infrastructure Programs, P40 OD010440) for providing strains. BG was supported by a grant from the NIH (R01 NS072129). MB is a Neuroscience Scholar of the Esther B O'Keeffe Charitable Foundation.

## Additional information

### Funding

| Funder | Grant reference number | Author |
|---|---|---|
| National Institute of Neurological Disorders and Stroke | R01 NS072129 | Brock Grill |

The funders had no role in study design, data collection and interpretation, or the decision to submit the work for publication.

### Author contributions

Melissa A Borgen, Conceptualization, Data curation, Formal analysis, Methodology, Writing—original draft; Andrew C Giles, Conceptualization, Data curation, Formal analysis, Investigation, Methodology, Writing—original draft; Dandan Wang, Formal analysis, Validation, Investigation, Methodology, Writing—original draft; Brock Grill, Conceptualization, Supervision, Funding acquisition, Writing—original draft, Project administration, Writing—review and editing

### Author ORCIDs

Melissa A Borgen http://orcid.org/0000-0003-2600-9946
Andrew C Giles http://orcid.org/0000-0002-0875-0696
Brock Grill https://orcid.org/0000-0002-0379-3267

### Decision letter and Author response

Decision letter https://doi.org/10.7554/eLife.44040.016
Author response https://doi.org/10.7554/eLife.44040.017

## Additional files

### Supplementary files
• Supplementary file 1. Transgenes and injection conditions.
DOI: https://doi.org/10.7554/eLife.44040.013
• Transparent reporting form
DOI: https://doi.org/10.7554/eLife.44040.014

### Data availability
All data generated or analysed during this study are included in the manuscript and supporting files.

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
