## [Decision Letter]

[Editors’ note: a previous version of this study was rejected after peer review, but the authors submitted for reconsideration. The first decision letter after peer review is shown below.]

Thank you for submitting your work entitled "Synapse maintenance is impacted by the RPM-1 signaling hub and the tubulin acetyltransferase ATAT-2" for consideration by *eLife*. Your article has been reviewed by two peer reviewers, and the evaluation has been overseen by a Reviewing Editor and a Senior Editor. The reviewers have opted to remain anonymous.

Our decision has been reached after consultation between the reviewers. Based on these discussions and the individual reviews below, we regret to inform you that your work will not be considered further for publication in *eLife*.

There is general agreement that the topic of synapse stability is important and of general interest. After decades of studying how synaptic connections are driven to form, there is new interest in what keeps them stable. As such, the topic should have a wide appeal in the field of neuroscience. Further, the reviewers generally agree that the genetic data, as presented, are solid and clear. Specifically, this refers to the identification of the new effects of *atat-2*. Further, it is acknowledged that the discovery of *atat-2* is important in the context of other microtubule regulatory genes that did not have a similar phenotype. Finally, the anti-correlation between growth cone and synapse was appreciated, but it was also felt that this was a side observation that, while interesting, did not much advance the core of the work.

With this enthusiasm in mind, there was also general agreement that the study relies heavily on a limited number of assays and would benefit greatly from a more in depth analysis of the microtubule cytoskeleton and the synapse. There are a number of studies that have pursued direct visualization of the microtubule cytoskeleton in the worm, either live or by EM analyses, and these have in some cases been performed in the context of synapse remodeling, which closely resembles the phenomenon being studied here. By directly addressing the phenotypes associated with the central argument of your work, it is expected that the interpretation of the genetics will be simplified and more powerful. There was general agreement that the behavioral analysis was difficult to interpret, both as a phenomenon and in terms of 'regulatory control' by the genes you are studying. Finally, there were concerns raised about prior work on the topic that seem directly relevant but not acknowledged.

Reviewer #1:

This manuscript (Borgen et al.) identifies the RPM-1 signaling protein and the tubulin acetyltransferase as key requirements for synapse maintenance and touch habituation.

The team first establishes that RPM-1 is required both for synapse stabilization and for the temporal coordination of synaptic bouton formation and axon termination in PLM neurons. RPM-1 localizes to presynaptic terminals just adjacent the active zone and mature active zone-bearing terminals are observed in *rpm-1* mutants prior to their destabilization. Next the team demonstrates that treatments that increase destabilization of microtubules appear to destabilize synapses. In alignment with these observations, the group shows that the tubulin acetyltransferase *atat-2* is required for synapse maintenance and that it interacts functionally with *rpm-1* via epistasis interactions, but other MT stabilizing proteins (e.g. Tau) do not appear to be involved. Finally, the group shows that disruption of the *rpm-1* decreases habituation to touch and that the effects of this mutation are much larger than the effects of *atat-2*.

Overall the data are clean and convincing. That said, this work is very descriptive – it defines new pieces involved in synapse maintenance but it does not provide new mechanistic insights into how these new players regulate synapse maintenance. As such, it would be much more appropriate for solid journal focusing on genetic mechanisms (e.g. Genetics).

1) The work describes new players but does not establish how these new players impact synapse stability. The lack of such a mechanistic understanding limits the impact of the discovery. One possibly productive area would be to measure microtubule dynamics in the various genetic backgrounds to test whether and how these treatments impact these. Alternatively, identifying key players or cargoes whose trafficking to synapses requires RPM-1 and/or ATAT-2 could provide mechanistic insights.

2) *rpm-1* and *atat-2* mutants have similar defects in PML synapse destabilization. The finding that habituation is compromised in *rpm-1* much more than in *atat-2* mutants would suggest that this phenotype is not simply explained by loss of these synapses. How do the authors reconcile this.

3) The finding that axon elongation and synapse formation are inversely coupled is very interesting observation but would be even more interesting if the authors could identify the underlying mechanisms.

Reviewer #2:

In this manuscript Borgen and colleagues describe a role for the signaling molecule Rpm-1 in microtubule-dependent synapse maintenance in *C. elegans*. Using the mechanosensory neuron PLM as a model system they demonstrate that the signaling molecule Rpm-1 is not required for synapse formation but for the control of synapse maintenance. Synaptic retraction phenotypes are shared by mutations in microtubule stabilizing factors including the α-tubulin acetyltransferase ATAT-2 and Mec-17 and can be modulated by pharmacological alterations of microtubule stability. Using genetic interaction assays the authors then try to establish a signaling pathway and propose that Rpm-1 acts via Atat-2 and in parallel to other microtubule stabilizing proteins to control synapse maintenance.

While this topic is in principle of great interest for the neuroscience community the current manuscript does not extend significantly enough beyond prior published work. It fails to sufficiently acknowledge prior work and does not incorporate or extend on these published results that addressed at least parts of this important question.

Three papers previously identified microtubule stability as a key factor for synapse maintenance using the same PLM synapse as a model system (Richardson et al., 2014 *eLife*; Marcette, Chen and Nonet, 2014 *eLife*; Chen et al., 2014). All three studies demonstrated that impairing MT stability by manipulating either Tubulin, Tubulin-binding proteins or Tubulin-modifying proteins results in defects in synapse maintenance. Importantly, these studies also demonstrated that pharmacological alterations mimic and/or modulate these effects, that synapse formation occurs normally and is followed by synapse retractions (e.g. Marcette Figure 5), that these effects are coupled to defects in the neurite extension and that *rpm-1, ptrn-1, mec-17* and *atat-2* are part of the signaling machinery controlling synapse maintenance. Furthermore, these studies demonstrated that defects in microtubule stability result in *dlk-1* activation that then in turn induces neuronal remodeling (neurite extension and synapse retraction). Mutations in *dlk-1* effectively suppress synaptic retraction phenotypes caused by impairments of microtubule stability. As Dlk-1 is the main effector of Rpm-1 (Nataka et al., 2005; *rpm-1* inhibits *dlk-1* activation) these studies already place *rpm-1* upstream of *dlk-1* in synapse maintenance (see Marcette, Chen and Nonet, 2014).

Thus the majority of phenotypes characterized in the Borgen et al. study have been previously described:

1) Rpm-1 required for synapse maintenance;

2) Coupling of neurite extension and retraction;

3) Peri-active zone localization of the *rpm-1*/Dlk-1 complex;

4) Time course of retraction phenotype for MT regulators (but not for *rpm-1*);

5) Pharmacological manipulation of MTs causing synaptic retractions;

6) Identification of *mec-17, ptrn-1* and *atat-2* as synapse stability genes).

In addition, there is convincing evidence that *dlk-1*upregulation is the main mediator of neuronal remodeling as a consequence of the MT perturbations. Rpm-1 as the main *dlk-1* inhibitor can thus clearly be integrated into this model (see Marcette, Chen and Nonet, 2014).

Unfortunately, Borgen et al., do not test and extend the current model but use (weak) genetic interaction data to suggest a reverse order of the signaling complex with *rpm-1* being upstream or parallel to the control of MT stability. In addition, the current manuscript does not provide sufficient evidence to support their proposed conclusions.

For these reasons I cannot recommend publication of this manuscript in *eLife*.

Additional comments:

1) Figure 1 there is no clear distinction between synapse retraction and axon branch degeneration. Throughout the paper it is not clear which comes first and all phenotype could be explained by axon degeneration (at 60h the axon is gone but remnants of the synapse remain).

In addition, there is a clear synapse formation phenotype. This should be quantified. Does the synaptic bouton region ever reach wild type dimensions? Impaired synapse formation might thus contribute to synaptic retraction.

2) Synapse retraction should be demonstrated using *unc-10* as a marker. Currently the authors only show that *unc-10* assembles at the synapse by 16 hr but do not demonstrate that *unc-10* (and thus the presynaptic active zone) is indeed disassembled in *rpm-1* mutants. This would also enable to differentiate between induced axonal degeneration and/or synaptic disassembly.

3) A large number of results have been previously published by Marcette, Chen and Nonet, 2014/Chen et al., 2014 – these findings are not acknowledged in this study (see general comment above).

4) The behavioral data is relatively meaningless – if these synapses are no longer present it is not surprising that the stimulus cannot evoke behavioral responses. It remains unclear why there are differences between *rpm-1* and *atat-2* mutants that display the same frequency of synaptic retractions.

5) The model is only based on genetic interactions and at least partly contradict findings from Marcette, Chen and Nonet, 2014, Borgen et al., 2017 and Chen et al., 2014. This should have been addressed experimentally.

---

## [Author Response]

[Editors’ note: the author responses to the first round of peer review follow.]

We appreciate the reviewers emphasizing the significance of our study’s topic, synapse maintenance. We also appreciate the helpful and thorough reviews we received. These comments have allowed us to dramatically overhaul and improve our paper, and have prompted numerous informative new experiments that address the reviewers’ concerns. We now highlight our novel finding that ATAT-2, acting via its enzymatic acetyltransferase activity, is required for synapse maintenance. As well as the novel discovery that ATAT-2 functions in a pathway with RPM-1 to regulate synapse maintenance.

Unfortunately, our prior paper did a poor job of highlighting these and other novel findings in our study. Our new manuscript was heavily rewritten to address this issue, and we now present new figures (Figure 6, and Figure 6—figure supplement 1) that show ATAT2 acetyltransferase activity regulates synapse maintenance (Figure 6B). We added many other new experiments and made several other changes to further increase the impact and novelty of our manuscript. These include:

1) A more detailed, thorough analysis of synapse maintenance defects allowed us to discover that subtle changes in the active zone marker UNC-10/RIM and presynaptic terminal size precede synapse destabilization in *rpm-1* (Figure 3G, H) and *atat-2* mutants (Figure 6—figure supplement 1B). These new findings add mechanistic insight into why synapse maintenance fails in *rpm-1* and *atat-2* mutants.

2) We add a new figure (Figure 3—figure supplement 1) showing that another active zone marker, SYD-2/Liprin, is present in *rpm-1* mutants during initial synapse assembly. This further strengthens the concept that synapse assembly is occurring in *rpm-1* mutants, and that synapse destabilization is the primary phenotype in the mechanosensory neurons of these animals.

3) We add numerous genetic experiments and show *rpm-1* is a hub in a genetic network that regulates synapse maintenance. Results indicate that ATAT-2 functions in the same pathway as RPM-1, but in parallel pathways to both PTRN-1 and DLK-1 (Figure 5C, D, E). Unexpectedly, our analysis also revealed that PTRN-1 functions in the same pathway as RPM-1 (Figure 5C).

4) Importantly, genetic results indicate that while *dlk-1* (lf) suppresses *rpm-1* (consistent with prior work), *dlk-1* fails to suppress *atat-2* (Figure 5E). This provides experimental support for the model that ATAT-2 functions downstream of RPM-1, and importantly indicates that ATAT-2 is a novel, DLK-1 independent mechanism RPM-1 utilizes to regulate synapse maintenance.

5) Our discovery that the acetyltransferase activity of ATAT-2 is required for synapse maintenance is a novel and particularly interesting observation for two reasons. First, synapse instability is a hallmark of many neurodegenerative diseases. Second, altered microtubule acetylation is associated with neurodegenerative diseases, such as Alzheimer’s and Parkinson’s.

We hope the reviewers agree that our extensive experimental and textual revisions now make our paper suitable for *eLife*. Below we detail specific responses to each reviewer’s individual comments.

[…] there was also general agreement that the study relies heavily on a limited number of assays and would benefit greatly from a more in depth analysis of the microtubule cytoskeleton and the synapse. There are a number of studies that have pursued direct visualization of the microtubule cytoskeleton in the worm, either live or by EM analyses, and these have in some cases been performed in the context of synapse remodeling, which closely resembles the phenomenon being studied here. By directly addressing the phenotypes associated with the central argument of your work, it is expected that the interpretation of the genetics will be simplified and more powerful. There was general agreement that the behavioral analysis was difficult to interpret, both as a phenomenon and in terms of 'regulatory control' by the genes you are studying. Finally, there were concerns raised about prior work on the topic that seem directly relevant but not acknowledged.Reviewer #1:[…] Overall the data are clean and convincing. That said, this work is very descriptive – it defines new pieces involved in synapse maintenance but it does not provide new mechanistic insights into how these new players regulate synapse maintenance. As such, it would be much more appropriate for solid journal focusing on genetic mechanisms (e.g. Genetics).

We appreciate the reviewer’s support for the significance and quality of our results. We agree that further experiments providing more mechanistic insight would be valuable. We detail changes to our manuscript and new experiments we have added to address these issues.

1) The work describes new players but does not establish how these new players impact synapse stability. The lack of such a mechanistic understanding limits the impact of the discovery. One possibly productive area would be to measure microtubule dynamics in the various genetic backgrounds to test whether and how these treatments impact these. Alternatively, identifying key players or cargoes whose trafficking to synapses requires RPM-1 and/or ATAT-2 could provide mechanistic insights.

We thank the reviewer for making an important point.

We attempted to analyze three different microtubule markers: EBP-2, PTRN-1, and UNC-104 in developing PLM neurons. We also attempted to analyze transport of RAB-3 in PLM neurons using time-lapse imaging. Unfortunately, our experiments were unsuccessful for a variety of reasons.

With regard to RAB-3, we actually found very little transport into and out of the presynaptic terminals in time frames examined. While somewhat unexpected, this suggests that the presynaptic terminals of PLM neurons might have limited or very slow trafficking that is difficult to detect with our existing reagents. In the future, we will consider trying FRAP to detect more subtle trafficking events into and out of presynaptic terminals.

With regard to EBP-2, PTRN-1 and UNC-104 markers, we were unable to get suitable expression with transgenic constructs for analysis. We incurred problems with detecting markers when single copy Mos insertion was used, and encountered many issues with the promoter we chose for traditional transgenic expression. Some of this could be due to our use of laser scanning confocal microscopy and might be improved with spinning disc confocal. However, our tri-institutional campus (Scripps Florida, Max Planck Florida and Florida Atlantic University) does not have a spinning disc microscope, which prevented us from testing this possibility in a reasonable timeframe. Thus, while the reviewer’s point is well taken, and a direction we hope to pursue further in the future, it could not be addressed as part of this study. Consistent with the difficulties we have encountered, we note that Nonet and colleagues made the following statement in their previous paper that examined the same mechanosensory neurons (Marcette, Chen and Nonet, 2014): “investigating neuronal microtubule dynamics in *C. elegans* neurons in vivo is technically difficult”.

As a result of these issues, we took a different direction in addressing the reviewer’s concern about mechanism. We engaged in a much more in depth developmental analysis of RAB-3 and UNC-10/RIM at the presynaptic terminals of wt, *rpm-1* and *atat-2* mutants. Our new results (Figure 3G, H) revealed subtle defects in the active zone and presynaptic bouton size of *rpm-1* mutants. These small, but significant defects precede synapse deterioration and provide a further explanation as to why synapse deterioration occurs in *rpm-1* mutants. Similar subtle defects in UNC-10 accumulation at presynaptic terminals of *atat-2* mutants were also observed at a critical time point just prior to synapse deterioration (Figure 6—figure supplement 1B).

Further, we add extensive new genetic analysis with *dlk-1* and *ptrn-1*. These experiments revealed that ATAT-2 functions in a linear pathway with RPM-1, but does so independently of DLK-1 and in parallel to PTRN-1. This indicates that the RPM1/ATAT-2 pathway regulates synapse maintenance in a DLK-1 independent manner, and provides experimental support for the concept that ATAT-2 is likely to function downstream of RPM-1. We comment on this point in both the Results and the Discussion in detail.

We now add further experiments and better present our novel discovery that ATAT-2 regulates synapse maintenance. Mechanistically, we show ATAT-2 functions via enzymatic tubulin acetyltransferase activity in the presynaptic mechanosensory neurons to regulate synapse maintenance (Figure 6, Figure 6—figure supplement 1).

Finally, more in depth analysis revealed that *atat-2* mutants, like *rpm-1* mutants, show subtle loss of the active zone marker UNC-10/RIM that precedes synapse destabilization (Figure 6—figure supplement 1).

2) rpm-1 and atat-2 mutants have similar defects in PML synapse destabilization. The finding that habituation is compromised in rpm-1 much more than in atat-2 mutants would suggest that this phenotype is not simply explained by loss of these synapses. How do the authors reconcile this.

The reviewer makes a fair point. We have updated the text to address this concern: “Prior studies (Crawley et al., 2017; Giles et al., 2015; Rankin and Wicks, 2000) and our results here are all consistent with impaired chemical synapses in mechanosensory neurons affecting habituation to repeated tap, a simple form of short-term learning. […] This idea is supported by previous studies in *Drosophila* which have shown that loss of function in the RPM-1 ortholog Highwire results in synapse formation defects and synaptic transmission defects that are mediated by distinct molecular mechanisms (Borgen et al., 2017a; Collins et al., 2006).”

3) The finding that axon elongation and synapse formation are inversely coupled is very interesting observation but would be even more interesting if the authors could identify the underlying mechanisms.

We agree with the reviewer. However, this data was presented a bit prematurely, and does not fit well with the focus of our new revised manuscript. This data also received conflicting reactions from our two reviewers. Therefore, we have removed this data from our revised manuscript. We hope to present this data with more mechanistic insight in the future.

We also opted to remove this data and not focus further on this part of our original paper because time was needed to address the extensive concerns raised by reviewer 2.

Reviewer #2:[…] While this topic is in principle of great interest for the neuroscience community the current manuscript does not extend significantly enough beyond prior published work. It fails to sufficiently acknowledge prior work and does not incorporate or extend on these published results that addressed at least parts of this important question.Three papers previously identified microtubule stability as a key factor for synapse maintenance using the same PLM synapse as a model system (Richardson et al., 2014; Marcette, Chen and Nonet, 2014; Chen et al., 2014). All three studies demonstrated that impairing MT stability by manipulating either Tubulin, Tubulin-binding proteins or Tubulin-modifying proteins results in defects in synapse maintenance. Importantly, these studies also demonstrated that pharmacological alterations mimic and/or modulate these effects, that synapse formation occurs normally and is followed by synapse retractions (e.g. Marcette Figure 5), that these effects are coupled to defects in the neurite extension and that rpm-1, ptrn-1, mec-17 and atat-2 are part of the signaling machinery controlling synapse maintenance. Furthermore, these studies demonstrated that defects in microtubule stability result in dlk-1 activation that then in turn induces neuronal remodeling (neurite extension and synapse retraction). Mutations in dlk-1 effectively suppress synaptic retraction phenotypes caused by impairments of microtubule stability. As Dlk-1 is the main effector of Rpm-1 (Nataka et al., 2005; rpm-1 inhibits dlk-1 activation) these studies already place rpm-1 upstream of dlk-1 in synapse maintenance (see Marcette, Chen and Nonet, 2014).

We thank the reviewer for their feedback. We appreciate the reviewer noting that the topic of our study is of “great interest to the neuroscience community”.

We have addressed the reviewer’s specific comments below extensively with both revisions to our text and new experiments. The result is a study that more comprehensively evaluates the relationship between *rpm-1* and a genetic network that affects synapse maintenance. Importantly, we now focus more on our novel discovery that ATAT-2 regulates synapse maintenance via its tubulin acetyltransferase activity. Unfortunately, our prior manuscript was not focused properly around this novel discovery. To further address this, we added several pieces of new data on ATAT-2 (Figures 5D, E, Figure 6, and Figure 6—figure supplement 1).

Our novel finding that ATAT-2 functions in the RPM-1 pathway (Figure 5) is now bolstered by new data showing that both *atat-2* and *rpm-1* mutants have subtle defects in accumulation of the active zone marker UNC-10/RIM that occurs just prior to synapse destabilization (Figure 3G and Figure 6—figure supplement 1).

Moreover, we now include new data on both PTRN-1 and DLK-1, which the reviewer notes were studied previously. The reviewer’s point, that our prior manuscript did not address the relationship between RPM-1 and ATAT-2 with regard to these important players, is well taken. Several interesting findings have resulted from our new experiments (Figure 5D, E). This has allowed us to paint a clearer, more comprehensive picture of how RPM-1 and ATAT-2 relate to PTRN-1 and DLK-1.

Our results indicate that RPM-1 and ATAT-2 tubulin acetyltransferase activity function in a linear genetic pathway presynaptically to regulate synapse maintenance (novel discovery). Genetic results indicate that PTRN-1 functions in the RPM-1 pathway (new finding), but acts in parallel to ATAT-2 (novel finding). Perhaps of particular interest to the reviewer is our finding that synapse maintenance defects in *atat-2* mutants are not suppressed by *dlk-1* (novel finding), although defects caused by *rpm-1* (lf) are suppressed by *dlk-1* (consistent with prior results). This indicates that ATAT-2 functions in the RPM-1 pathway, but independently of DLK-1. As the reviewer notes DLK-1 is a prominent mechanism of RPM-1 function, our results show that ATAT-2 is an entirely new mechanism of RPM-1 function that does not rely upon DLK-1. Our results also provide evidence suggesting that ATAT-2 functions downstream of RPM-1 since we would expect *atat-2* to be suppressed by *dlk-1* if it functioned upstream of RPM-1.

Importantly, it is not unreasonable that RPM-1 interacts genetically with several molecules other than DLK-1, as numerous studies in *C. elegans* have now established that RPM-1 utilizes 6 different downstream mechanisms (GLO-4, PPM-2, ANC-1, RAE1, MLK-1 and DLK-1; for review see Grill et al., 2016). While DLK-1 is a prominent player, it is not the only player in the RPM-1 signaling network.

The reviewer’s comments prompted us to re-evaluate and substantially overhaul our entire manuscript, as well as add an extensive body of new experiments. We think this has dramatically improved the novelty and impact of our paper, which we hope the reviewer agrees is suitable for publication in *eLife*.

Below we detail our specific experimental and textual updates to this manuscript.

Thus the majority of phenotypes characterized in the Borgen et al. study have been previously described:1) Rpm-1 required for synapse maintenance;

The reviewer makes a fair point that a previous study (which we cite) described a role for *rpm-1* in synapse formation (Schaefer, et al., 2000). However, the data from this study are more consistent with impaired synapse assembly in *rpm-1* mutants and mild effects on synapse maintenance (See Schaefer et al., 2000, Figure 3).

Importantly, this prior study by the Nonet group only examined the presence or absence of presynaptic boutons.

Our study expands extensively from this prior work in several ways. 1) Our results differ with the findings of Schaefer et al., as we now show that the principle defect in *rpm-1* mutants is failed synapse maintenance rather than impaired synapse assembly. 2) We do this using developmental time course analysis for not only bouton morphology (two transgenes tested Figure 1 and Figure 3), but also three different presynaptic proteins marking synaptic vesicles (RAB-3, Figure 3), and the active zone (UNC-10/RIM, Figure 3; and SYD-3/Liprin, Figure 3—figure supplement 1). None of these markers were analyzed in *rpm-1* mutants in the original Nonet study, and none were tested over developmental time course previously. 3) We perform new quantitative analysis on bouton size and the presence of UNC-10 at presynaptic boutons in *rpm-1* mutants during development, and show that at the critical 16 hour PH time point *rpm-1* mutants have small defects in bouton size and active zone accumulation (Figure 3G, H). Thus, subtle presynaptic defects occur at a key time point just before major loss of synapse stability in *rpm-1* mutants. This is also a novel dataset that significantly informs our mechanistic understanding of why synapse maintenance fails in *rpm-1* mutants.

2) Coupling of neurite extension and retraction;

We think there is some confusion about our presentation of growth cone frequency with synaptic bouton frequency. Both reviewers have commented on this data with differing positive and negative reactions. Since this is not at the core of discoveries made in this paper, we have removed this data from our revised paper.

We hope to present this data in the future more clearly and with better mechanistic insight.

3) Peri-active zone localization of the rpm-1/Dlk-1 complex;

We have adjusted the text to more accurately reflect our own analysis (done in mechanosensory neurons for the first time) with previous studies that focused on motor neurons (Abrams et al., 2008).

“In adult animals, RPM-1 localized directly adjacent to UNC-10 at presynaptic terminals (Figure 2C). This indicates RPM-1 localizes to the periactive zone of presynaptic terminals in mechanosensory neurons. Our observation is consistent with prior studies that examined RPM-1 localization in motor neurons (Abrams et al., 2008).”

4) Time course of retraction phenotype for MT regulators (but not for rpm-1);

We agree with the reviewer that we did a quantitative time course of presynaptic bouton morphology and RAB-3 presence in *rpm-1* mutants, but more quantitation for *rpm-1* mutants would be valuable. We now include new quantitative developmental time course data on bouton size, and UNC-10 accumulation in *rpm-1* mutants (Figure 3G, H). We also include new data on quantitation of SYD-2 active marker in *rpm-1* mutants (Figure 3—figure supplement 1).

Given the focus of our revised manuscript on our novel discovery that ATAT-2

acetyltransferase activity regulates synapse maintenance, we opted to also perform more in depth analysis of synapse maintenance defects in *atat-2* mutants (Figure 6, Figure 6—figure supplement 1). This showed that UNC-10 and RAB-3 accumulate at presynaptic terminals of *atat-2* mutants, but by adulthood synapses are lost. Moreover, more in depth analysis showed that subtle UNC-10 defects occur at a small number of synapses in *atat-2* mutants (Figure 6—figure supplement 1B). This is strikingly similar to what we observe in *rpm-1* mutants (Figure 3G).

Thus, we now add an extensive amount of new data to address this concern.

5) Pharmacological manipulation of MTs causing synaptic retractions;

We agree with the reviewer that this phenomenon has been shown previously, which we cite in the Results section.

“Consistent with prior work (Chen et al., 2014; Richardson et al., 2014), treating wt animals with colchicine, a microtubule-destabilizing drug, resulted in loss of PLM synapses (Figure 4A).”

However, the relationship between colchicine, taxol and *rpm-1* loss of function has not been evaluated previously for synaptic defects in any system. Notably, we did do this type of experiment with regard to axon termination (Borgen, Wang and Grill, 2017). We now show our data for the effects of these drugs on *rpm-1* mutants in the context of synapse maintenance (Figure 4). However, it is notable that our findings on the relationship between *rpm-1* (lf) and these drugs is the opposite of what occurs in the context of axon termination (Borgen, Wang and Grill, 2017). Thus, between our prior study (Borgen, Wang and Grill, 2017) and this new study we unveil a surprising finding: The relationship between RPM-1 and microtubules varies with the phenotype and where it occurs in the axon (axon termination versus synapse maintenance) even though both phenotypes were evaluated in the same neurons.

To add further novelty, we add new experiments showing colchicine treatment of *atat-2* mutants (Figure 6C). Our results on RPM-1 (Figure 4) and ATAT-2 (Figure 6C) indicate RPM-1 and ATAT-2 have similar relationships to pharmacological manipulation of microtubules in the context of synapse maintenance. This is consistent with another novel discovery we present, that RPM-1 and ATAT-2 function in the same pathway to regulate synapse maintenance.

Importantly, the reviewer’s comments helped us realize a major problem with our prior paper is that it was written to focus much too heavily on microtubule effects, and neglected to properly emphasize our novel discoveries regarding synapse maintenance. To address this, we have removed conclusive strong statements regarding RPM-1, ATAT-2 and microtubules from our paper and instead focus on several novel discoveries including that ATAT-2 regulates synapse maintenance via its acetyltransferase activity, that ATAT-2 functions in the RPM-1 pathway to regulate presynaptic maintenance, and that ATAT-2 functions independently of DLK-1 to regulate synapse maintenance.

6) Identification of mec-17, ptrn-1 and atat-2 as synapse stability genes).

We agree that *ptrn-1*, which is a microtubule minus end binding protein, has been shown to regulate synapse maintenance (Marcette, Chen and Nonet, 2014). Indeed, we cite this paper numerous times in our revised manuscript.

Marcette and colleagues do look at *mec-17* and *atat-2* mutants in their study. However, it is important to emphasize they only examined axon termination defects (neurite overgrowth defects) and did not analyze synaptic defects. It is important to point this out as Marcette and colleagues use the term “neurite remodeling”. This can lead to confusion, since it could imply that both axon termination and synapse formation were evaluated, when this is not the case.

Moreover, Marcette et al. do not explore whether ATAT-2 or MEC-17 regulate synapse assembly or maintenance in their paper. Nor do they explore the genetic relationship between *rpm-1* and *atat-2* or *mec-17*. Notably, they did examine *rpm-1; ptrn-1* double mutants, but conclude these genes are enhancers. We now add new data (Figure 5C), which shows that *rpm-1* and *ptrn-1* function in the same pathway (i.e.synapse maintenance defects are not significantly increased in *rpm-1; ptrn-1* double mutants compared to *rpm-1* single mutants).

As noted above, we also add numerous new experiments on ATAT-2 showing it regulates synapse maintenance using multiple synaptic markers, and quantitative analysis (Figure 6, Figure 6—figure supplement 1). Most importantly, we show that ATAT-2 functions via its tubulin acetyltransferase activity to regulate synapse maintenance (Figure 6A, B).

To further increase novelty, we add new experiments on *ptrn-1; atat-2* double mutants. Our results indicate the frequency of synapse maintenance defects is enhanced in these double mutants. The genetic relationship between *ptrn-1* and *atat-2* has not been assessed previously.

In summary, we have added numerous experiments that increase novelty and substantially expand on prior work. We have also paid careful attention to citing prior work and ensuring we are fair and accurate in the presentation of our findings that are novel, and well as differences with prior work.

In addition, there is convincing evidence that dlk-1 upregulation is the main mediator of neuronal remodeling as a consequence of the MT perturbations. Rpm-1 as the main dlk-1 inhibitor can thus clearly be integrated into this model (see Marcette, Chen and Nonet, 2014).

We thank the reviewer for highlighting the importance of addressing DLK-1 in our study. We agree that DLK-1 is an important mediator, although not the only mediator, of RPM-1 function (Grill et al., 2016).

We now test the genetic relationship between *atat-2* and *dlk-1*. Interestingly, while *atat-2* is in the *rpm-1* pathway (Figure 5C), *atat-2* is not dependent upon *dlk-1 (i.e. atat-2; dlk-1* double mutants are not suppressed while *rpm-1; dlk-1* double mutants are suppressed, see Figure 5E).

This is genetic evidence that ATAT-2 acts independently of DLK-1. It is also consistent with ATAT-2 functioning downstream of RPM-1. This new experiment was extremely valuable and provided further novelty by showing the RPM-1/ATAT-2 pathway is a new, DLK-1 independent mechanism for regulating synapse maintenance.

Unfortunately, Borgen et al., do not test and extend the current model but use (weak) genetic interaction data to suggest a reverse order of the signaling complex with rpm-1 being upstream or parallel to the control of MT stability. In addition, the current manuscript does not provide sufficient evidence to support their proposed conclusions.

We aren’t certain what the reviewer is referring to, but we suspect they disagree with our model that RPM-1 most likely functions upstream of ATAT-2. While we thought this was most likely, it is a fair point that our original paper lacked data to directly address this model.

We now show new data indicating that synapse maintenance defects in *atat-2* mutants are not suppressed by *dlk-1* (Figure 5E). In contrast, defects in *rpm-1* mutants are suppressed by *dlk-1* (Figure 5E). Given these findings and evidence that *rpm-1* and *atat2* function in the same pathway (Figure 5C), the simplest model that explains our results is that ATAT-2 and DLK-1 are part of a signaling network that is differentially regulated downstream of RPM-1 (summarized in revised Figure 8). Because *dlk-1* suppresses *rpm-1* but not *atat-2*, it is particularly likely that ATAT-2 functions downstream of RPM-1. If this were not the case, one would expect suppression of both *rpm-1* and *atat-2* by *dlk1*, which did not occur.

To try and provide further evidence that RPM-1 functions upstream of ATAT-2, we attempted transgenic bypass experiments in which ATAT-2 was overexpressed in *rpm-1* mutants. Unfortunately, despite extensive efforts on this front our experiments were not conclusive. However, these experiments could have failed for many reasons and negative outcomes here do not invalidate our model. Although, positive outcomes would have made us more confident.

To further address the reviewer’s concern, we have substantially revised our summary figure (Figure 8).

We have also updated our Discussion to handle this issue carefully and note the caveat to our model:

“Our results do not provide definitive evidence for the order of ATAT-2 and RPM-1 within this novel pathway. […] Despite cumulative reasons for favoring the model that RPM-1 functions upstream of ATAT-2, we cannot entirely rule out the alternative possibility.”

We ask the reviewer to note that it is not unreasonable some regulators of microtubules could lie upstream of RPM-1 while others, such as ATAT-2, function downstream. Indeed, our prior work (Borgen et al., 2017) suggests that PTL-1/Tau is a potential upstream inhibitor of RPM-1 in the context of axon termination, but data here indicate a different relationship between RPM-1 and PTL-1/Tau during synapse maintenance. Given that our Discussion is already quite lengthy, more in depth commentary on these ideas is beyond the scope of this paper. However, we hope to incorporate a discussion on these ideas in a review article in the future.

For these reasons I cannot recommend publication of this manuscript in eLife.Additional comments:1) Figure 1 there is no clear distinction between synapse retraction and axon branch degeneration. Throughout the paper it is not clear which comes first and all phenotype could be explained by axon degeneration (at 60h the axon is gone but remnants of the synapse remain).

The reviewer notes that a bouton is present at 60hr (Figure 1), but this is the bouton from the PLM on the back side of the animal. A synaptic branch is present, but is outside our confocal imaging depth. We have updated our schematic in the figure and the legend to make this clear. =

We note that during the developmental time when maintenance fails (between 16 and 48 hours PH), we commonly observe synaptic branches that lack boutons. In contrast, we never observe presynaptic boutons that lack synaptic branches. Thus, it is unlikely synaptic branch degeneration is triggering synapse loss. On this particular point we are in agreement with the Nonet group: when synapse maintenance is impaired the synaptic branch retracts (Marcette, Chen and Nonet, 2014, Luo et al., 2014).

In addition, there is a clear synapse formation phenotype. This should be quantified. Does the synaptic bouton region ever reach wild type dimensions? Impaired synapse formation might thus contribute to synaptic retraction.

The reviewer makes a good point. We have significantly expanded our analysis by quantitatively assessing both presynaptic bouton size and the presence of UNC-10 at 12 and 16 hours PH. We now show that *rpm-1* synapses start out normal in size and are similar to wt initially (5 and 7hr PH, Figure 3H). Synapses in *rpm-1* mutants accumulate UNC-10 at normal frequency through 12 hours PH (Figure 3G). However, UNC-10 levels begin to drop a small but significant amount just prior to synapse destabilization at 16 hours PH (Figure 3G). Thus, just prior to branch loss, we observe small reductions in bouton size (Figure 3H) and the UNC-10 active zone marker (Figure 3G). These results indicate that while synapses are assembling, they begin to show subtle abnormalities corresponding with onset of synapse destabilization.

To further address this concern, we now add a third presynaptic marker, SYD-2/liprin, to our analysis (Figure 3—figure supplement 1). In *rpm-1* mutants, SYD-2 accumulates at presynaptic terminals normally prior synapse destabilization.

2) Synapse retraction should be demonstrated using unc-10 as a marker. Currently the authors only show that unc-10 assembles at the synapse by 16 hr but do not demonstrate that unc-10 (and thus the presynaptic active zone) is indeed disassembled in rpm-1 mutants. This would also enable to differentiate between induced axonal degeneration and/or synaptic disassembly.

We have addressed this with new data showing UNC-10 quantitation at 12 and 16hr PH (Figure 3G). Our data suggest that UNC-10 is present at all presynaptic terminals in *rpm-1* mutants at 12 hours PH, and then subtle defects in UNC-10 accumulation occur at 16 hours PH, the key time point just prior to loss of synapses. As noted above, this data is consistent with synapse assembly being largely normal in *rpm-1* mutants, and subtle abnormalities in presynaptic terminals occurring just before the onset of synapse destabilization which culminates in total loss of the presynaptic terminal and the synaptic branch. Above we have also discussed why it is unlikely that axon degeneration is leading to synapse loss.

3) A large number of results have been previously published by Marcette, Chen and Nonet, 2014/Chen et al., 2014 – these findings are not acknowledged in this study (see general comment above).

We appreciate the reviewer’s concern. We cited both of these studies and their findings in our initial manuscript. We reread both these papers front to back, and pay very careful attention to citing these prior studies in our revised paper.

Further, we have removed the prior emphasis on microtubule stability/dynamics, and instead focus on synapse maintenance in our revised manuscript. Importantly, both reviewers agreed that synapse maintenance is a very important topic in neuroscience.

Below we detail where we now cite these papers. We must note, that some of our results do not agree with the Marcette paper. We have tried to note this with careful, reasonable language. Indeed, this is a delicate issue, as we greatly respect the tremendous contributions Mike Nonet and his lab have made to understanding synapse formation.

Chen at al. and Marcette et al. are cited in the Introduction:

“For instance, pharmacological and genetic perturbation of microtubules impairs presynaptic bouton maintenance in these cells (Chen et al., 2014). Genetic screens using mechanosensory neurons revealed that the microtubule minus end binding protein PTRN-1/CAMSAP and the actin binding protein ZYX-1 are required for synapse maintenance (Luo et al., 2014; Marcette et al., 2014; Richardson et al., 2014).”

Chen at al. is cited in the Results:

“Consistent with prior work (Chen et al., 2014; Richardson et al., 2014), treating wt animals with colchicine, a microtubule-destabilizing drug, resulted in loss of PLM synapses (Figure 4A).”

Marcette et al., is cited in the Results:

“Our observation that ptrn-1 affects PLM synapse maintenance is consistent with a prior study (Marcette et al., 2014).”

Due to differing results with *rpm-1; ptrn-1* double mutants compared to Marcette et al., we also note this discrepancy in the Results:

“the frequency of synapse maintenance defects was not increased in rpm-1; ptrn-1 double mutants compared to single mutants, which suggests PTRN-1 and RPM-1 function in the same pathway (Figure 5C). We note that this result differs with a prior study that suggested RPM-1 and PTRN-1 function in parallel pathways to regulate synapse formation (Marcette, Chen and Nonet, 2014).”

4) The behavioral data is relatively meaningless – if these synapses are no longer present it is not surprising that the stimulus cannot evoke behavioral responses. It remains unclear why there are differences between rpm-1 and atat-2 mutants that display the same frequency of synaptic retractions.

Reviewer 1 also asked for clarification on the behavioral habituation experiments. We addressed this by rewriting our Results on habituation much more carefully and clearly.

We also specifically note the following statement and explanation:

“Prior studies (Crawley et al., 2017; Giles et al., 2015; Rankin and Wicks, 2000) and our results here are all consistent with impaired chemical synapses in mechanosensory neurons affecting habituation to repeated tap, a simple form of short-term learning. […] This idea is supported by previous studies in *Drosophila* which have shown that loss of function in the RPM-1 ortholog Highwire results in synapse formation defects and synaptic transmission defects that are mediated by distinct molecular mechanisms (Borgen et al., 2017a; Collins et al., 2006).”

We think this data remains valuable because it provides independent behavioral evidence that RPM-1 and ATAT-2 function in the same pathway. We now emphasize this point, rather than focusing on the link between synaptic effects in mechanosensory neurons and behavioral habituation to repeated tap.

5) The model is only based on genetic interactions and at least partly contradict findings from Marcette, Chen and Nonet, 2014, Borgen et al., 2017 and Chen et al., 2014. This should have been addressed experimentally.

We agree that our genetic results differ with some findings from Marcette, Chen and Nonet, 2014. We note this in our revised paper.

As noted above, we now provide new data on *ptrn-1, atat-2, dlk-1* and their genetic relationships with *rpm-1* and one another. This has resulted in a much more novel and comprehensive study focused on molecular genetic mechanisms regulating synapse maintenance.

Our findings agree with Marcette and colleagues that *ptrn-1* mutants have defects in synapse maintenance. However, our results disagree about *ptrn-1* functioning in parallel to *rpm-1*, as mentioned above. Importantly, we note that all our experiments were done after 4X outcrossing to wt (*N2*) and show that *ptrn-1* is epistatic to *rpm-1* instead of functioning in parallel. We feel it’s important to share these results with the community.

With respect to our prior study, we were equally surprised that the genetic relationship between *rpm-1* and mutants that affect microtubule stability differ based on the phenotype and subcellular location examined. We think it is critical to report our unexpected, but important finding to the community. We think it is plausible that a molecule such as RPM-1 that functions as a signaling hub to regulate axon termination and synapse formation (Grill, et al., 2016) could be a sophisticated regulator of signaling that influences microtubule stability with differing outcomes in different axonal compartments. Indeed, our own recent paper argues that effects of RPM-1 on axon termination can be mechanistically distinguished from synapse formation/maintenance via the MIG-15/*JNK*-1 pathway (Crawley et al., 2017).

In the Discussion we attempt to address this more clearly and note its importance for the field with the following paragraph:

“It is intriguing that the functional genetic relationship between RPM-1 and molecules that affect microtubule stability differs between synapse maintenance in mechanosensory neurons described here (Figure 8), and axon termination of the same neurons that occurs in a different anatomical location (Borgen et al., 2017b). […] These findings in *C. elegans* might explain why studies in fish and mice that analyzed different types of neurons arrived at opposing conclusions about how PHR protein signaling influences microtubule stability (Hendricks and Jesuthasan, 2009; Lewcock et al., 2007).”